# Genome-wide screens identify *Toxoplasma gondii* determinants of parasite fitness in IFNγ-activated murine macrophages

Yifan Wang 1,8, Lamba Omar Sangaré1,8, Tatiana C. Paredes-Santos1, Musa A. Hassan 2,3,4, Shruthi Krishnamurthy1, Anna M. Furuta1, Benedikt M. Markus 5,6, Sebastian Lourido 5,7 & Jeroen P. J. Saeij 1✉

Macrophages play an essential role in the early immune response against *Toxoplasma* and are the cell type preferentially infected by the parasite in vivo. Interferon gamma (IFNγ) elicits a variety of anti-*Toxoplasma* activities in macrophages. Using a genome-wide CRISPR screen we identify 353 *Toxoplasma* genes that determine parasite fitness in naïve or IFNγ-activated murine macrophages, seven of which are further confirmed. We show that one of these genes encodes dense granule protein GRA45, which has a chaperone-like domain, is critical for correct localization of GRAs into the PVM and secretion of GRA effectors into the host cytoplasm. Parasites lacking GRA45 are more susceptible to IFNγ-mediated growth inhibition and have reduced virulence in mice. Together, we identify and characterize an important chaperone-like GRA in *Toxoplasma* and provide a resource for the community to further explore the function of *Toxoplasma* genes that determine fitness in IFNγ-activated macrophages.

¹ Department of Pathology, Microbiology & Immunology, School of Veterinary Medicine, University of California, Davis, Davis, CA, USA. ² College of Medicine and Veterinary Medicine, The University of Edinburgh, Edinburgh, UK. ³ The Roslin Institute, The University of Edinburgh, Edinburgh, UK. ⁴ Center for Tropical Livestock Health and Genetics, The University of Edinburgh, Edinburgh, UK. ⁵ Whitehead Institute for Biomedical Research, Cambridge, MA, USA. ⁶ Faculty of Biology, University of Freiburg, Freiburg, Germany. ⁷ Department of Biology, Massachusetts Institute of Technology, Cambridge, MA, USA. ⁸These authors contributed equally: Yifan Wang, Lamba Omar Sangaré. ✉email: jsaeij@ucdavis.edu

Toxoplasma gondii is an obligate intracellular parasite that causes disease in immunocompromised individuals, such as HIV/AIDS patients, and when contracted congenitally[1]. It causes lifelong infections by converting from rapidly dividing tachyzoite stages into encysted slow-growing bradyzoites, which mainly localize to the brain and muscle tissues. Once established in the host cell, Toxoplasma resides in a unique replication niche called the parasitophorous vacuole (PV), which is separated from the host cytoplasm by the PV membrane (PVM) and does not fuse with the endolysosomal system[2,3].

The cytokine interferon gamma (IFNγ) is essential for the control of Toxoplasma replication in host cells[4]. Mice with macrophages that can no longer respond to IFNγ are extremely susceptible to Toxoplasma demonstrating their crucial role in IFNγ-mediated control of Toxoplasma[5]. IFNγ-induced upregulation of immunity-related GTPases (IRGs) and guanylate binding proteins (GBPs)[6–8] are key mechanisms for controlling Toxoplasma in mice[9–13]. IRGs and GBPs can bind and vesiculate the PVM, eventually leading to the death of the parasite[14–17]. In addition, IFNγ induces the production of nitric oxide (NO) and reactive oxygen species (ROS), which lead to parasite damage and growth restriction in murine macrophages[18]. IFNγ stimulation also changes the metabolism of cells thereby affecting the availability of nutrients to pathogens[19]. For example, IFNγ-activated cells can limit the availability of iron, L-tryptophan, cholesterol, polyamines, and dNTPs[20]. The potency of IFNγ-induced anti-Toxoplasma activities in murine macrophages is parasite-strain-dependent, with the clonal type II and III strains being more susceptible to IFNγ-induced growth restriction compared to type I strains[21].

Toxoplasma co-opts host cells by secreting ROP and GRA effector proteins, from organelles called rhoptries and dense granules, respectively, into the host cell or onto the PVM (reviewed in[22]). ROP18 (encoding a secreted kinase)[23,24], and ROP5 (encoding a member of an expanded family of pseudokinases)[25,26] are highly polymorphic and, together with ROP17[27] and GRA7[28,29], account for differences in parasite virulence in mice by cooperatively blocking IRGs[14,30–32] and GBPs[6,33] loading on the PVM. Other parasite-strain-specific effectors include ROP16 and GRA15, which can affect GBP1 loading on the PVM[34] and macrophage polarization into the classical (M1) or alternative (M2) activation phenotypes[35]. GRA15 was recently shown to recruit IRGB6 and GBP1-5 to the PVM via interacting with TRAF6, thereby enhancing the susceptibility to IFNγ-induced parasite elimination[36]. The pseudokinase ROP54 also inhibits GBP2 loading on the PVM[37]. Additional secreted effectors, such as GRA12, affect Toxoplasma survival in murine macrophages without affecting the loading of IRGB6 to the PVM[38]. Besides targeting IFNγ-induced host cell effectors, Toxoplasma can directly inhibit IFNγ signaling. For example, through a GRA (TgIST) secreted beyond the PVM[39,40], Toxoplasma can directly inhibit STAT1 transcriptional activity[41–43], an essential component of the IFNγ signaling pathway[44].

The secretion of GRAs beyond the Toxoplasma PVM into the host cell cytosol is dependent on the MYR complex[45–47], which might form a PVM-localized translocon. A Golgi-resident aspartyl protease ASP5, which cleaves the Toxoplasma export element (TEXEL, RRLxx) motif in some GRAs, is required for exported GRAs to reach the host cytosol and also for the correct localization of other GRAs confined to the PV[48–50]. How GRA effectors that eventually insert into the PVM are kept from inserting into other membranes during their traffic from the ER via the Golgi to the dense granules and after their secretion into the PV lumen is largely unknown. It has been hypothesized that specific chaperones in the dense granules and in the PV lumen could bind to these hydrophobic GRA effectors and aid their traffic to the correct destination[51,52]. Because PVM-localized GRAs are at the host-parasite interface they determine the acquisition of host-derived nutrients, can mediate the evasion of the host cytosolic immune response, and form the components of the translocon that secretes other GRAs into the host cytoplasm. It is therefore likely that these PVM-localized GRAs and the putative chaperones that mediate the correct trafficking of these GRAs to the PVM will be important for parasite fitness in IFNγ-activated cells.

Many Toxoplasma genes determining parasite survival in IFNγ-activated murine cells were initially identified using genetic crosses between strains that differ in virulence in mice. However, this approach fails to identify strain-independent genes. Here we used a genome-wide loss-of-function screen in the type I (RH) strain, which is resistant to IRGs/GBPs-mediated killing, to identify 160 Toxoplasma genes that determine fitness in IFNγ-activated murine macrophages and 193 genes that determine fitness in naïve macrophages. Seven of the high-confidence hits were confirmed by testing single-gene knockouts in functional parasite growth assays in IFNγ-activated murine macrophages. We demonstrate that GRA45, one of these hits, has a chaperone-like function and plays an important role in the trafficking and localization of GRA effectors to the PVM and the secretion of GRA effectors beyond the PVM. Δgra45 parasites have enhanced susceptibility to IFNγ-mediated growth inhibition in murine, rat, and human macrophages, and significantly reduced virulence in mice. These results provide functional information to the community to investigate the mechanism by which other Toxoplasma genes affect fitness in naïve or IFNγ-activated macrophages.

## Results

**A loss-of-function screen identifies Toxoplasma genes that determine fitness in naïve and IFNγ-activated murine macrophages.** We previously identified Toxoplasma genes that determine parasite fitness during infection of human foreskin fibroblasts (HFFs) using a genome-wide loss-of-function CRISPR/Cas9 screen[53]. However, it is likely that this set of fitness-conferring genes varies by host cell type, host species, and after infection of IFNγ-activated cells. The challenge in Toxoplasma genome-wide loss-of-function screens is to maintain the complexity of the mutant pool during the selection. It was previously reported that macrophages require stimulation by IFNγ along with LPS or TNFα to effectively restrict Toxoplasma growth[54]. Such stimulated macrophages are extremely effective in inhibiting Toxoplasma growth—even of virulent strains like the type I (RH) strain—which would likely create a bottleneck during the selection of the mutant parasite pool. IFNγ stimulation by itself was sufficient for a modest but significant inhibition of Toxoplasma growth in C57BL/6 J murine bone-marrow-derived macrophages (BMDMs) (Supplementary Fig. 1a). By contrast, IFNγ + TNFα stimulation drastically restricted parasite growth (Supplementary Fig. 1b). Thus, we reasoned that a loss-of-function screen in macrophages stimulated with IFNγ alone would reduce the random loss of mutants while, at the same time, allowing us to identify mutants that are susceptible to IFNγ-mediated growth restriction. A Toxoplasma mutant pool was generated by transfecting RH parasites constitutively expressing Cas9 (RH-Cas9) with a library of sgRNAs containing ten guides for each of the 8156 Toxoplasma genes[53]. This mutant pool was passaged in HFFs to remove mutants with general fitness defects (e.g., invasion and replication) before growing in naïve and IFNγ-activated BMDMs for three passages (Fig. 1a, b). Continuous passages in HFFs served as a control to identify mutants with fitness defects in naïve BMDMs but not in HFFs. The relative abundance of sgRNAs was measured using Illumina sequencing

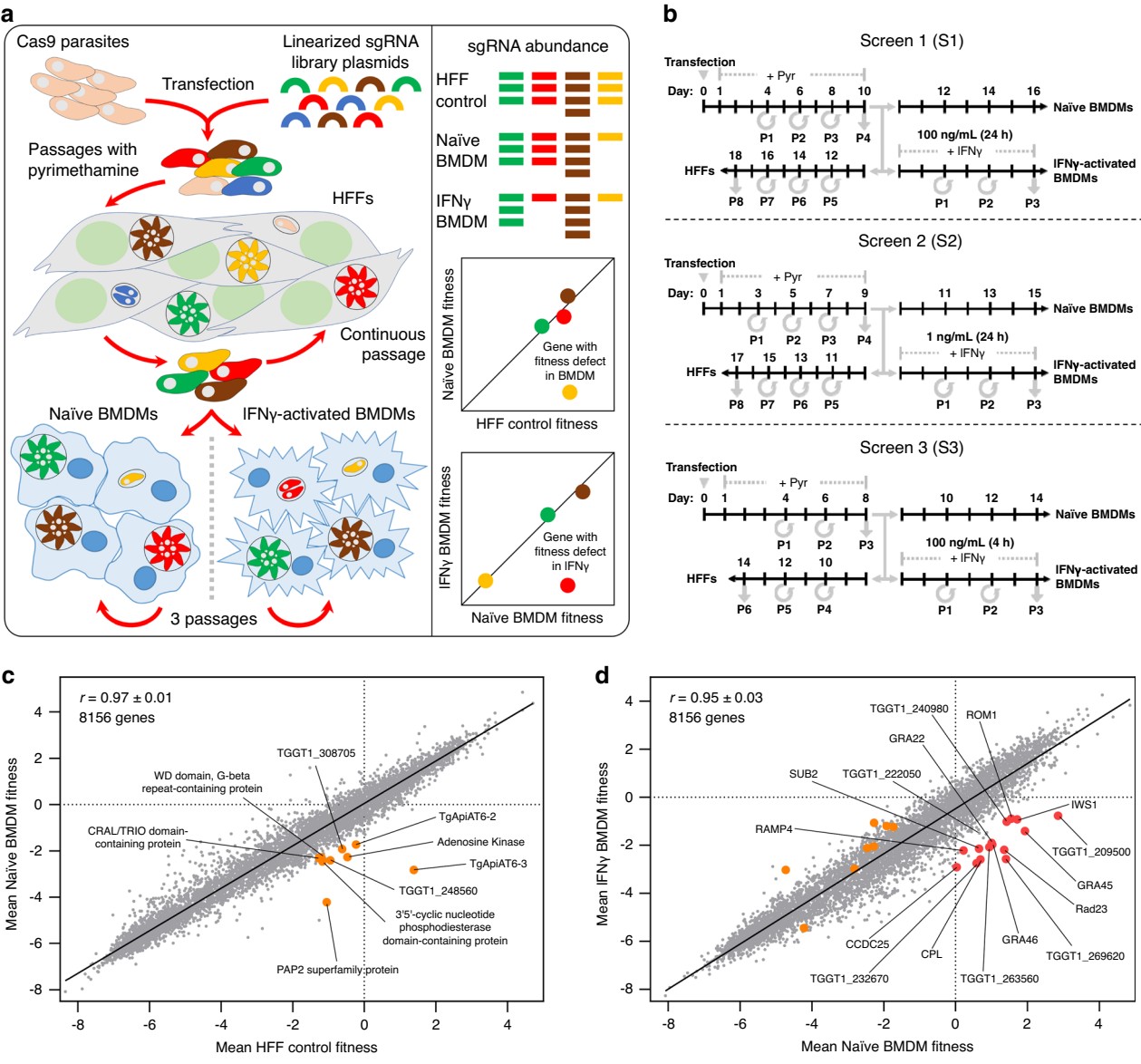

**Fig. 1 *Toxoplasma gondii* genome-wide loss-of-function screen in naïve or IFNγ-activated murine bone-marrow-derived macrophages. a** Screening workflow. At least $5 \times 10^8$ Cas9-expressing RH parasites were transfected with linearized plasmids containing 10 sgRNAs against every *Toxoplasma* gene. Transfected parasites were passaged in HFFs under pyrimethamine selection to remove non-transfected parasites and parasites that integrated plasmids with sgRNAs targeting parasite genes important for fitness in HFFs. Subsequently, the pool of mutant parasites was either continuously passaged in HFFs or passaged for three rounds in murine BMDMs that were left unstimulated or prestimulated with IFNγ. The sgRNA abundance at different passages, determined by illumina sequencing, was used for calculating fitness scores and identifying genes that confer fitness in naïve or IFNγ-activated BMDMs. **b** Timeline for the generation of mutant populations and subsequent selection in the presence or absence of murine IFNγ. Times at which parasites were passaged (P) are indicated. **c** Correlation between mean parasite gene fitness scores in Naïve BMDMs and HFF control. Orange dots indicate nine high-confidence candidate genes that confer fitness in naïve BMDMs compared to HFFs (Table 1). **d** Correlation between mean parasite gene fitness scores in IFNγ-activated and Naïve BMDMs. Red dots indicate 16 high-confidence IFNγ fitness-conferring candidate genes in murine BMDMs and orange dots the nine high-confidence genes conferring fitness in naïve BMDMs compared to HFFs (Table 1).

from the input library and the parasite DNA isolated from the early passage in HFFs (prior to BMDM infection), three additional passages on naïve (referred to as Naïve BMDM) or IFNγ-activated BMDMs (referred to as IFNγ BMDM), and the late passage in HFFs (referred to as HFF control). We calculated the average $\log_2$ fold change in abundance of sgRNAs targeting a specific gene in the samples (early passage HFF, Naïve BMDM, IFNγ BMDM, or HFF control) relative to the input library and defined it as the "fitness" score for that gene (Supplementary Data 1). The fitness scores of genes after early passage in HFFs were highly correlated ($r = 0.81 \pm 0.03$, mean ± SEM, $n = 3$) with

previously determined HFFs "phenotype" scores[53], highlighting the reproducibility of these screens and demonstrating that up to the point of infecting the BMDMs, the mutant pool was behaving as expected.

Because the intracellular environments between murine BMDMs and HFFs are likely quite different, certain *Toxoplasma* genes could be important for fitness in naïve BMDMs but not HFFs. To identify this set of genes, we determined the probability of being fitness conferring in naïve BMDMs for each gene by comparing the fitness (Naïve BMDM vs. HFF control) distribution to those of 497 control genes from our previous study[55].

After removing the genes not expressed in murine macrophages[56], we identified 193 parasite genes (Supplementary Fig. 3b), including nine high-confidence hits (Fig. 1c and Table. 1), with a significant negative enrichment ($p < 0.05$) and a large effect size (Cohen's $d ≥ 0.8$) compared to control genes (Supplementary Data 1). To determine if these *Toxoplasma* genes shared biological annotation we performed gene set enrichment analysis (GSEA)[57] and Gene Ontology (GO) enrichment analysis (ToxoDB.org). We found significant enrichment of *Toxoplasma* genes involved in folate metabolism, purine metabolism, and parasite membrane-localized transporters (Supplementary Data 2), suggesting that nutrient availability to *Toxoplasma* in naïve BMDMs and HFFs is variable.

To identify parasite genes that determine fitness in IFNγ-activated BMDMs, we performed three independent screens with minor changes to the IFNγ concentration and time of stimulation: 100 ng/mL of IFNγ for 24 h (referred to as screen 1 or S1), 1 ng/mL of IFNγ for 24 h (referred to as screen 2 or S2), and 100 ng/mL of IFNγ for 4 h (referred to as screen 3 or S3; Fig. 1b). A lower IFNγ concentration (1 ng/mL) induced less parasite growth inhibition (Supplementary Fig. 1a). In addition, IFNγ (100 ng/mL) stimulation of murine BMDMs for 4 h induced a similar but slightly lower expression of gene sets that had ≥2-fold upregulation in IFNγ stimulation for 24 h (Supplementary Fig. 2a, b, and Supplementary Data 3). The screens with a lower IFNγ concentration (S2) or with the high concentration for less time (S3) indeed reduced the bottleneck and associated stochastic loss of mutants (Supplementary Fig. 3a). Next, by subtracting the Naïve BMDM fitness score from the IFNγ BMDM fitness score we identified 160 parasite genes with expression in murine macrophages that specifically determine fitness in IFNγ-activated BMDMs ($p < 0.05$) with a large effect size (Cohen's $d ≥ 0.8$) (Supplementary Fig. 3c and Supplementary Data 1). GO analysis showed that genes encoding for proteins with peptidase activity or DNA-binding/transcription regulator activity were significantly enriched (Supplementary Data 2). In astion, we performed pathway analysis on the RNA-seq data to identify pathways that are uniquely regulated by IFNγ in the murine macrophages. The cholesterol biosynthesis pathway (Supplementary Fig. 2c, d) and many other metabolic pathways were significantly downregulated in IFNγ-activated BMDMs compared to naïve BMDMs (Supplementary Data 3). *Toxoplasma* genes involved in lipid biosynthesis/transport were enriched in the set of genes that determined fitness in IFNγ-activated BMDM vs. naïve BMDM (Supplementary Data 2), suggesting that they are required for responding to changes in cholesterol levels in IFNγ-activated BMDMs[58]. However, *ROP18* and *GRA12*, two *Toxoplasma* genes determining RH resistance to IFNγ in murine macrophages[30,38], were not present as candidates likely due to the differences between the individual screen conditions although they had a 2.2-fold ($p = 0.08$ and Cohen's $d = 0.65$) and 5.2-fold ($p = 0.07$ and Cohen's $d = 1.64$) lower fitness in IFNγ-activated BMDMs, respectively. It should be noted that ROP5 was not in our list of hits because the sgRNA design algorithm excluded ROP5-targeting sgRNAs because no uniquely targeting sgRNAs could be designed as ROP5 is a multi-copy gene. Among all the genes conferring fitness in IFNγ-activated BMDMs, 16 high-confidence candidate genes (Fig. 1d) are presented in Table 1. ROM1 (TGGT1_200290) and SUB2 proteases (TGGT1_314500) are important for microneme proteins (MICs) processing and rhoptry protein maturation, respectively[59,60], possibly indicating the role of specific MICs and ROPs in the invasion of and survival in IFNγ-activated macrophages. Cathepsin CPL (TGGT1_321530) is important for the digestion of host cytosolic proteins in the parasite lysosomal-like organelle called the vacuolar compartment[61]. Four dense granule proteins GRA22 (TGGT1_215220), GRA45

(TGGT1_315620), GRA46 (TGGT1_208370), and a putative GRA (TGGT1_263560; Supplementary Fig. 5i) were present as high-confidence hits. There are three putative DNA repair-like proteins (TGGT1_269620, TGGT1_209500, and TGGT1_295340); three proteins involved in intracellular trafficking, putative Cytohesin-3 (TGGT1_232670), putative cytoplasmic dynein (TGGT1_240980), and putative ribosome associated membrane protein RAMP4 (TGGT1_276940); one protein involved in metabolism, putative methylcitrate synthase (TGGT1_222050); CCDC25 (TGGT1_228300) and putative IWS1 transcription factor (TGGT1_227560). Taken together, we identified multiple *Toxoplasma* candidate genes that confer fitness specifically in naïve or IFNγ-activated BMDM.

**Single parasite gene knockouts from hits identified by CRISPR screens have fitness defects in IFNγ-activated BMDMs.** To confirm some of the parasite genes that determine fitness in IFNγ-activated BMDMs, we generated individual knockouts of four genes (TGGT1_269620, GRA45, TGGT1_232670, and TGGT1_263560) in luciferase-expressing RH parasites (Supplementary Fig. 4a, d–f). All knockouts had reduced growth compared to wild-type parasites in IFNγ-activated BMDMs as determined by luciferase assays at 24 h post-infection (p.i.) (Fig. 2a–d). We previously determined the contribution to the in vivo fitness of 217 *Toxoplasma* genes[55]. Of the 31 genes that determined in vivo fitness, Δ*gra22* and Δ*TGGT1_269950* parasites were outcompeted by wild-type in the peritoneum[55]. GRA22 and TGGT1_269950 have average fitness scores $< -2$ in IFNγ-activated *vs.* Naïve BMDMs (Table 1) suggesting that the in vivo fitness of these genes is likely due to their importance in determining parasite fitness in IFNγ-activated peritoneal macrophages. To confirm this, we tested the growth of Δ*gra22*, Δ*TGGT1_269950* and wild-type parasites in IFNγ-activated BMDMs and found that IFNγ significantly inhibited Δ*gra22* and Δ*TGGT1_269950* parasite growth relative to wild-type parasites (Fig. 2e, f). Most of the knockouts had no, or a minor, fitness defect in naïve murine BMDMs (Supplementary Fig. 5a–f) or mouse embryonic fibroblasts (MEFs) (Supplementary Fig. 5g) confirming that these genes are specific for conferring fitness in the presence of IFNγ. Δ*TGGT1_263560* and Δ*gra22* parasites formed larger plaques compared to wild-type parasites in MEFs, although this was only significant for Δ*TGGT1_263560* (Supplementary Fig. 5g). Although the Δ*TGGT1_263560* plaques were larger, they appeared quite different compared to wild-type plaques; the lysis area contained very few parasites and host cells, possibly due to an early egress phenotype (Supplementary Fig. 5h). To test the impact of candidate gene knockouts in a non-luciferase-expressing type I (RH) strain, we performed a growth competition assay, in which growth differences between wild-type and knockouts accumulate over three passages in both naïve and IFNγ-activated BMDMs. Δ*gra22* parasites were outcompeted by wild-type parasites only in the IFNγ-activated BMDMs but not naïve BMDMs (Fig. 2g). Similar to Δ*gra22* parasites, parasites lacking another high-confidence hit, *ROM1*, (Supplementary Fig. 4h) were outcompeted by wild-type parasites in IFNγ-activated BMDMs (Fig. 2h). Thus, the high-confidence parasite genes identified by our genome-wide loss-of-function screen indeed determine parasite fitness in IFNγ-activated BMDM.

**GRA45 is important for parasite fitness in IFNγ-stimulated macrophages of other species.** Of the seven genes we confirmed to have a fitness defect in IFNγ-activated BMDM, four genes (GRA22, GRA45, TGGT1_263560, and TGGT1_269950) encode known or putative secretory proteins with a predicted signal peptide (ToxoDB.org). GRA22 was previously characterized and

**Table 1 *Toxoplasma* genes that determine fitness in naïve or IFNγ-activated macrophages.**

**Genes with a fitness defect in naïve macrophages**

| Gene ID | Product description | Naïve BMDM vs. HFF Control fitness (mean ± SEM) | p value (one-sided Wilcoxon signed-rank test) | Cohen's d | IFNγ vs. Naïve BMDM fitness | In vitro phenotype in HFFs[53] | In vivo fitness[55,84] |
|---|---|---|---|---|---|---|---|
| TGGT1_249580 | *TgApiAT6-3* | −4.2 ± 1.1 | 0.001 | 7.6 | −0.2 | 0.5 | N/A |
| TGGT1_247360 | PAP2 superfamily protein | −3.2 ± 1.0 | 0.002 | 5.8 | −1.2 | 1.6 | N/A |
| TGGT1_250880 | Kinase, pfkB family protein (Adenosine Kinase) | −1.8 ± 0.4 | 0.002 | 3.4 | 1.2 | 1.4 | N/A |
| TGGT1_290860 | *TgApiAT6-2* | −1.5 ± 1.2 | 0.03 | 2.8 | 0.5 | 1.0 | N/A |
| TGGT1_248560 | Hypothetical protein; homology to 26 S Proteasome regulatory subunit RPN2 (Swiss-Model) | −1.5 ± 0.5 | 0.003 | 2.8 | 0.1 | 0.3 | N/A |
| TGGT1_308075 | Hypothetical protein; homology to DNAse Pyocin AP41 (Swiss-Model) | −1.3 ± 0.2 | 0.003 | 2.5 | 0.7 | 1.5 | N/A |
| TGGT1_224840 | 3′5′-cyclic nucleotide phosphodiesterase domain-containing protein | −1.3 ± 0.3 | 0.004 | 2.4 | 0.3 | 0.4 | N/A |
| TGGT1_262650 | WD domain, G-beta repeat-containing protein | −1.1 ± 0.4 | 0.007 | 2.1 | 0.2 | 0.4 | N/A |
| TGGT1_269390 | CRAL/TRIO domain-containing protein | −1.0 ± 0.6 | 0.03 | 2.0 | −0.1 | 0.1 | N/A |

**Genes with a fitness defect in IFNγ-activated macrophages**

| Gene ID | Product description | IFNγ vs. naïve BMDM fitness (mean ± SEM) | p value (one-sided Wilcoxon signed-rank test) | Cohen's d | Naïve BMDM vs. HFF control Fitness | In vitro phenotype in HFFs[53] | In vivo fitness[55,84] |
|---|---|---|---|---|---|---|---|
| TGGT1_269620 | DNA repair protein-like (Swiss-model) | −4.0 ± 1.6 | 0.004 | 2.9 | 1.5 | −0.3 | N/A |
| TGGT1_209500 | DNA repair protein-like (HHpred) | −3.6 ± 1.8 | 0.006 | 2.7 | 0.4 | 0.1 | N/A |
| TGGT1_295340 | UV excision repair protein Rad23 protein | −3.5 ± 1.0 | 0.003 | 2.6 | 1.4 | −0.4 | N/A |
| TGGT1_316250 | GRA45 | −3.3 ± 0.4 | 0.002 | 2.4 | 1.0 | 1.2 | N/A |
| TGGT1_232670 | Cytohesin-3-like; GEF of Arf6 (Swiss-Model) | −3.3 ± 0.8 | 0.003 | 2.4 | 0.3 | 1.5 | N/A |
| TGGT1_321530 | Cathepsin L (CPL) | −3.3 ± 0.9 | 0.003 | 2.4 | 1.7 | 0.7 | N/A |
| TGGT1_263560 | Putative GRA (Supplementary Fig. 5i) | −3.0 ± 1.2 | 0.006 | 2.2 | 1.5 | 0.5 | N/A |
| TGGT1_208370 | GRA46 | −3.0 ± 1.6 | 0.008 | 2.1 | 1.5 | 1.8 | N/A |
| TGGT1_228300 | CCDC25 protein | −2.9 ± 0.6 | 0.003 | 2.1 | 0.3 | −1.1 | N/A |
| TGGT1_215220 | GRA22 | −2.9 ± 1.8 | 0.02 | 2.1 | 1.1 | 1.0 | Peritoneum |
| TGGT1_314500 | SUB2 | −2.8 ± 1.4 | 0.01 | 2.0 | 0.8 | 0.3 | N/A |
| TGGT1_227560 | Putative IWS1 transcription factor | −2.7 ± 1.7 | 0.02 | 1.8 | 1.3 | −0.7 | N/A |
| TGGT1_200290 | ROM1 | −2.4 ± 1.3 | 0.01 | 1.7 | 0.2 | 0.8 | N/A |
| TGGT1_240980 | Cytoplasmic dynein-like (Swiss-Model/HHpred) | −2.4 ± 0.9 | 0.007 | 1.7 | 0.3 | −1.0 | N/A |

**Table 1 (continued)**

Genes with a fitness defect in IFNγ-activated macrophages

| Gene ID | Product description | IFNγ vs. naïve BMDM fitness (mean ± SEM) | p value (one-sided Wilcoxon signed-rank test) | Cohen's d | Naïve BMDM vs. HFF control Fitness | In vitro phenotype in HFFs[53] | In vivo fitness[55,84] |
|---|---|---|---|---|---|---|---|
| TGGT1_276940 | Putative ribosome associated membrane protein RAMP4 | −2.4 ± 1.4 | 0.02 | 1.7 | −0.4 | 0.4 | N/A |
| TGGT1_222050 | Putative methylcitrate synthase (Swiss-Model) | −2.2 ± 1.3 | 0.03 | 1.5 | 0.6 | 0.7 | N/A |
| TGGT1_269950[a] | Putative GRA (ToxoDB.org); homology to QSOX (HHpred) | −2.1 ± 1.2 | 0.2 | 1.4 | 1.4 | 0.4 | Peritoneum |

Upper panel: nine top hits, selected from 193 naïve BMDM fitness-conferring genes, with at least four sgRNAs present in the last passage of HFF control and ≥2-fold lower fitness (fitness score ≤ −1) in naïve BMDMs vs. HFF control. Genes with a mean Naïve BMDM fitness > −1.5 and a mean HFF control fitness < −1.5 were removed. Lower panel: 16 top hits (not including TGGT1_269950), selected from 232 IFNγ fitness-conferring genes, with at least four sgRNAs present in the 3rd passage of naïve BMDMs and a ≥4-fold lower fitness (fitness score ≤ −2) in IFNγ vs. naïve BMDMs. Genes with a mean IFNγ fitness >−0.5 and a mean naïve BMDM fitness <0 were removed. The genes are ranked according to the mean fitness (naïve BMDM vs. HFF control, upper panel; IFNγ vs. naïve BMDM, lower panel). See Supplementary Data 1 for the complete list.
[a]TGGT1_269950 was confirmed as an IFNγ fitness-conferring gene (Fig. 2f), although it does not fit the above criteria.

shown to contribute to parasite natural egress[62], while GRA45 was recently discovered as an ASP5 substrate with unknown function[49]. TGGT1_263560 and TGGT1_269950 were annotated as hypothetical proteins (ToxoDB.org) without previous characterization. We observed that TGGT1_263560 colocalized with PV-resident GRA2 in intracellular parasites (Supplementary Fig. 5i) and considered it as a putative GRA. TGGT1_269950 was predicted as a putative GRA based on published dense granule proteomic analysis[63] and hyperLOPIT spatial proteomic analysis[64] (ToxoDB.org). IRG-mediated destruction of the PVM is one of the main mechanisms by which murine cells inhibit *Toxoplasma* growth. We therefore determined the IRGA6/IRGB6 coating on the vacuole of these knockouts (Fig. 3a, b). We observed slightly increased loading of IRGA6 (Fig. 3a), and a significantly larger fraction of Δ*TGGT1_263560* and Δ*TGGT1_269950* vacuoles with IRGB6 coating, but no difference in Δ*gra45* and Δ*gra22* parasites (Fig. 3b). This indicates that GRA45 and GRA22 employ different mechanisms to counter IFNγ-mediated parasite growth inhibition in murine macrophages.

Macrophages from different species have different mechanisms to control *Toxoplasma* growth[8] and therefore knowing if these genes are important for conferring resistance to IFNγ in other species could help us narrow down the possible mechanism by which they do so. To test if these four GRAs are also important for parasite fitness in IFNγ-activated macrophages of other species, primary rat BMDMs and human THP-1 macrophages were stimulated with IFNγ or left unstimulated followed by infection with wild-type, Δ*gra45*, Δ*gra22*, Δ*TGGT1_263560*, or Δ*TGGT1_269950* parasites (Fig. 3c–h). Similar to what was observed in naïve murine BMDMs, these knockouts had no, or a minor but not significant, fitness defects in naïve rat BMDMs and human THP-1 macrophages (Supplementary Fig. 6). Although Δ*gra22* parasites had significantly higher growth reduction in primary rat BMDMs activated with IFNγ (5 ng/mL) (Fig. 3c), Δ*gra45* parasites were the only knockout that was significantly more susceptible to IFNγ-mediated growth inhibition compared to wild-type parasites in both rat and human macrophages (Fig. 3c–h) indicating GRA45 plays an important function in the resistance of *Toxoplasma* to IFNγ in multiple species. We therefore decided to focus on determining the function of GRA45.

**GRA45 has structural homology to protein chaperones and maintains the correct localization of GRAs to the PVM.** GRA45 was recently shown to interact with GRA44 (TGGT1_221870) and With-No-Gly-loop (WNG)2 kinase (TGGT1_240090), which also localize to dense granules[49]. GRA45, GRA44, and WNG2 have orthologues in coccidian species belonging to the Sarcocystidae (*Neospora caninum*, *Hammondia hammondi*, *Cystoisospora suis*, *Sarcocystis neurona*) with >40% amino acid identities and Eimeriidae (*Eimeria* spp. and *Cyclospora cayetanensis*) with >30% amino acid identities (ToxoDB.org). This is a rather unique phylogenetic profile as most GRAs are only conserved within the *Toxoplasmatinae* suggesting that these GRAs have a conserved function in these different parasite species. In addition, a GRA45 paralogue (TGGT1_295390) is exclusively expressed in cat intestinal stages, compared to GRA45, which is highly expressed in tachyzoite and bradyzoite stages and has moderate expression levels in sexual stages (ToxoDB.org). Searches for primary sequence homology or known protein domains failed to provide any suggestions for putative GRA45 functions. However, homology searches based on secondary-structure prediction indicated that the N-terminal region of GRA45 (from amino acids 72–225) has structural homology to the α-Crystallin domain (ACD) of small heat shock proteins (sHSPs) (Supplementary

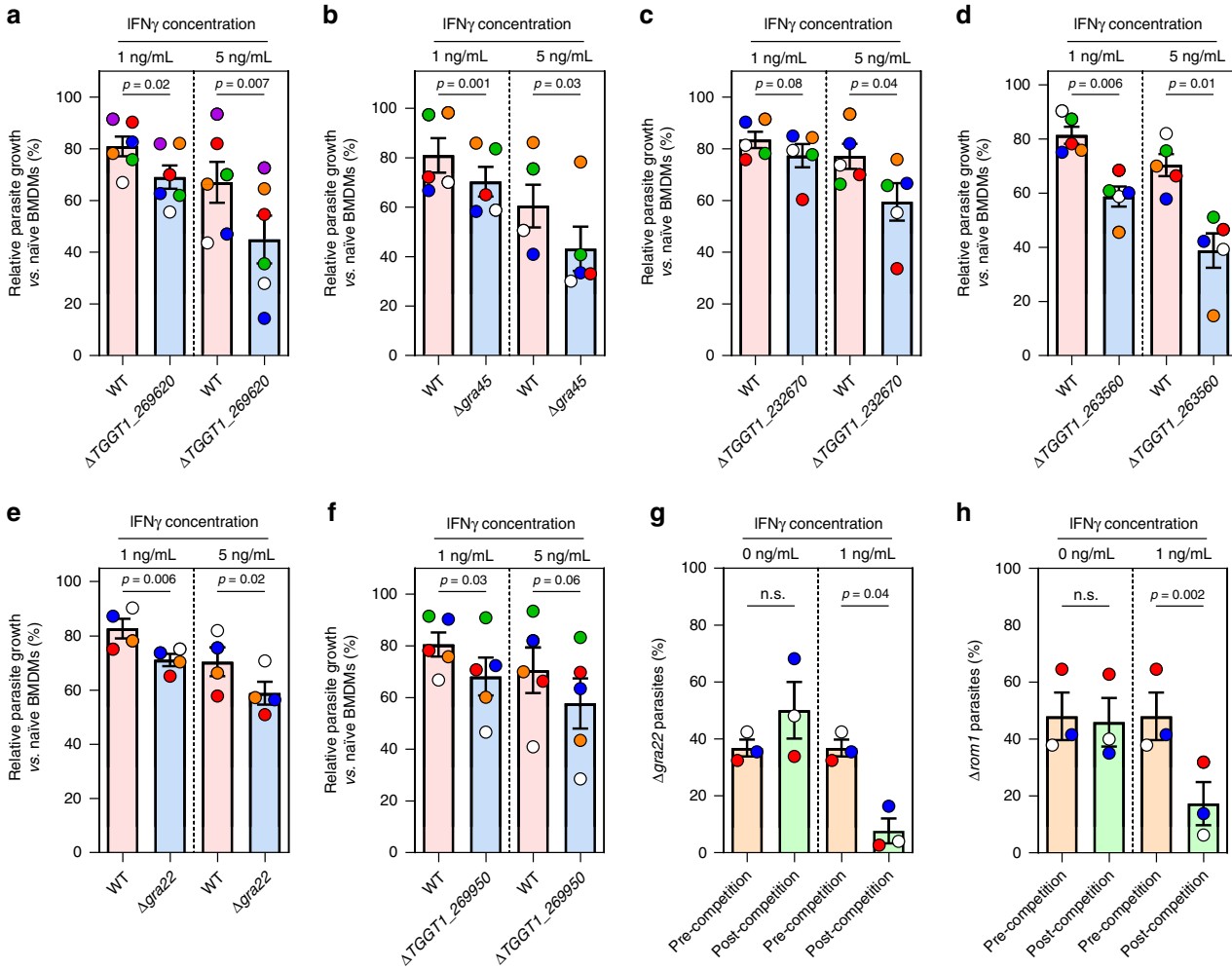

**Fig. 2 Validation of candidate genes that determine parasite fitness in IFNγ-stimulated murine BMDMs. a–f** Murine BMDMs prestimulated with IFNγ (1 or 5 ng/mL) or left unstimulated for 24 h were infected with luciferase-expressing wild-type (WT) parasites or with parasites in which *TGGT1_269620* (**a**, n = 6), *GRA45* (**b**, n = 5), *TGGT1_232670* (**c**, n = 5), *TGGT1_263560* (**d**, n = 5), *GRA22* (**e**, n = 4), or *TGGT1_269950* (**f**, n = 5) was knocked out (MOI of 0.25). Parasite growth for each strain was measured by luciferase assay at 24 h p.i. Parasite growth in IFNγ-activated BMDMs is expressed relative to growth in naïve BMDMs. Data are displayed as mean ± SEM with independent experiments indicated by the same color dots. The significant difference between WT and knockout was analyzed with two-tailed paired *t*-test. **g, h** Growth competition assay between GFP-positive WT parasites and GFP-negative **g** Δ*gra22* or **h** Δ*rom1* parasites was performed in murine BMDMs prestimulated with 1 ng/mL IFNγ or left unstimulated for three passages. The percentage of Δ*gra22* or Δ*rom1* was determined at the start of the competition and after three passages by plaque assay measuring the GFP-negative plaques vs. total plaques. Data are displayed as mean ± SEM with independent experiments (n = 3) indicated by the same color dots. The significant difference between the ratio of WT vs. knockout before and after competition was analyzed with two-tailed paired *t*-test.

Fig. 7a), which are ATP-independent chaperones that maintain the native folding status of proteins under stress conditions. The conserved I/VxI/V motifs present in 90% of sHSPs was conserved in GRA45 and most of its orthologues (Supplementary Fig. 7a). The most significant structural homology was to Aggregation Suppressing Protein A (AgsA) from *Salmonella typhimurium*[65] and HSP20 from *Xylella fastidiosa*[66] (Supplementary Fig. 7a). The GRA45 C-terminal region (amino acids 260–335) had structural slogy to DUF1812 domains[67] and Mfa2[68], which share a transthyretin-like fold (Supplementary Fig. 7a). Transthyretin family members are known to bind and transport lipids such as retinol and thyroxine[69]. In addition, the top 10 models (TM-score 0.9–0.69) with GRA45 structural homology, as predicted by I-TASSER[70], are predicted to bind hydrophobic substrates as they have acyltransferase domains. Overall these data indicate GRA45 harbors a potential chaperone-like function with preference for hydrophobic substrates. Consistent with published results that GRA45 remains in the PV lumen after secretion[49], we observed

that almost all GRA45 colocalized with PV-resident GRA1 and GRA2 in the PV lumen but only partially with PVM-associated GRA5 and GRA7, even though GRA45 colocalized with all these GRAs to dense granules in extracellular parasites (Supplementary Fig. 7b). Thus, GRA45 is unlikely to be a *Toxoplasma* effector that directly interacts with host proteins to modulate or inhibit IFNγ-induced anti-*Toxoplasma* activities.

Based on its structural homology to sHSPs, we hypothesized that instead of directly neutralizing IFNγ-mediated inhibition, GRA45 might perform a chaperone-like function for other GRA effectors, of which some are known to counteract the murine IFNγ response[28,29,38]. GRAs with a hydrophobic domain (such as a transmembrane domain or amphipathic α-helices) have peculiar trafficking and do not traffic to the parasite plasma membrane but instead traffic to the dense granules where they exist in partially soluble and aggregated states[51]. What prevents the aggregation of these hydrophobic domain-containing proteins in dense granules or insertion of their transmembrane domain

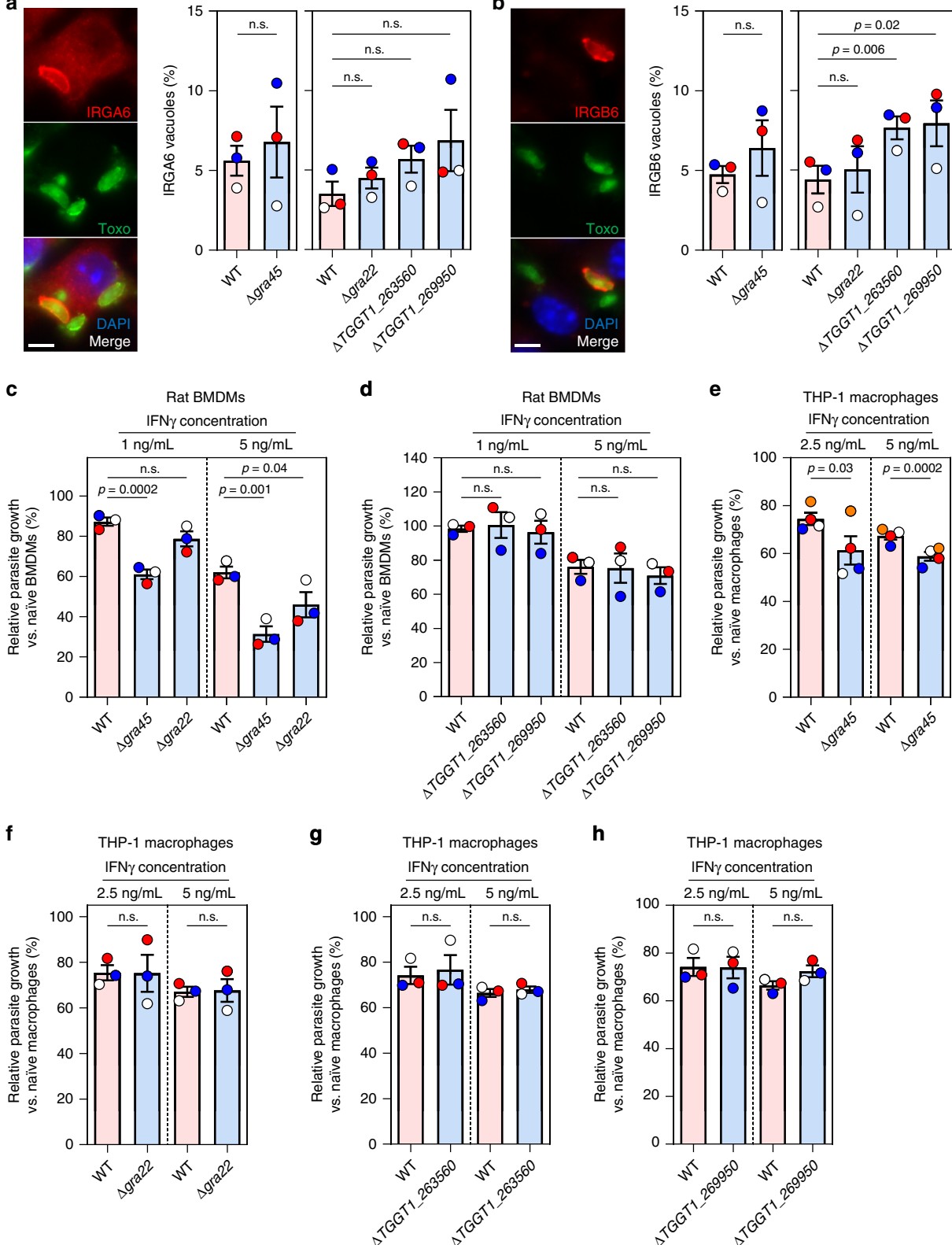

into the ER membrane or the dense granule membrane is unknown but has been hypothesized to be mediated by dense granule protein chaperone(s)[51,52]. To test the hypothesis that GRA45 is a dense granule protein chaperone, we first examined the solubility of hydrophobic GRA2, which has two amphipathic helices that mediate its membrane association post secretion[71]. In parasite lysates generated by freeze/thawing, GRA2 was in both

pellet and supernatant fractions in wild-type, $\Delta gra45$, and $\Delta gra45$ parasites complemented with a C-terminal HA-tagged version of $GRA45$ ($\Delta gra45 + GRA45HA$, Supplementary Fig. 4b, d) and ran according to its corresponding molecular weight (MW) (Fig. 4a). In addition to its predicted size, GRA2 appeared at a much higher MW in the pellet fraction from all strains suggesting insoluble protein aggregates as the bands were not completely separated

Fig. 3 Δgra45 parasites have enhanced susceptibility to IFNγ-mediated growth inhibition in rat BMDMs and human THP-1 macrophages. a, b Murine BMDMs prestimulated with 5 ng/mL of IFNγ for 24 h were infected with luciferase-expressing WT, Δgra45, Δgra22, ΔTGGT1_263560, or ΔTGGT1_269950 parasites (MOI of 0.5) for 1 h and subsequently fixed, permeabilized, and stained for IRGA6 (a) or IRGB6 (b). A representative image from three independent experiments is shown on the left (Scale bar = 5 µm). Quantification of IRGA6/IRGB6 loading on at least 200 vacuoles is presented on the right panel. Data are displayed as mean ± SEM with independent experiments indicated by the same color dots. The significant difference between WT and knockout was analyzed with two-tailed paired t-test. c, d Brown Norway rat BMDMs prestimulated with or without IFNγ (1 ng/mL or 5 ng/mL) were infected with luciferase-expressing WT, Δgra45 (c), Δgra22 (c), ΔTGGT1_263560 (d), or ΔTGGT1_269950 (d) parasites (MOI of 0.25) for 24 h. Parasite growth for each strain was measured by luciferase assay and the growth in IFNγ-activated BMDMs is expressed relative to growth in naïve BMDMs. Data are displayed as mean ± SEM with independent experiments (n = 3) indicated by the same color dots. The significant difference between WT and knockout was analyzed with two-tailed paired t-test. e–h PMA-differentiated THP-1 macrophages prestimulated with or without IFNγ (2.5 or 5 ng/mL) were infected with luciferase-expressing WT, Δgra45 (e, n = 4), Δgra22 (f, n = 3), ΔTGGT1_263560 (g, n = 3), or ΔTGGT1_269950 (h, n = 3) parasites (MOI of 0.25) for 24 h. Parasite growth for each strain was measured by luciferase assay and the growth in IFNγ-activated THP-1 macrophages is expressed relative to growth in naïve THP-1 macrophages. Data are displayed as mean ± SEM with independent experiments indicated by the same color dots. The significant difference between WT and knockout analyzed with two-tailed paired t-test.

during the gel migration. As a control, we used the parasite plasma membrane protein SAG1, which traffics to the parasite surface independent of the dense granules trafficking pathway. No high MW bands of SAG1 were observed in any of the parasite strains (Supplementary Fig. 10) indicating that the high MW bands of GRA2 were specific to dense granule proteins. After treatment with non-ionic detergents (NP-40), which solubilize membrane proteins by associating with their hydrophobic surface[51], monomeric GRA2 was completely solubilized and released into the supernatant fraction in all the strains (Fig. 4a). Although the high MW bands were also solubilized after NP-40 treatment, consistently less solubilization of GRA2 high MW bands was observed in Δgra45 parasites compared to wild-type and Δgra45 + GRA45HA parasites (Fig. 4a and Supplementary Fig. 8a). We then observed the solubility of GRA7, which contains a transmembrane domain but localizes to the dense granule core and does not integrate into the dense granule membrane[72,73]. In all strains, GRA7 mainly presented in the pellet fraction with both monomeric bands and high MW bands (Fig. 4b). Similar to GRA2, high MW GRA7 in Δgra45 parasites was more resistant to NP-40-induced solubilization (Fig. 4b); however, because of variability between experiments there was no significant difference between wild-type and Δgra45 parasites (Supplementary Fig. 8b). No high MW bands or monomeric GRA7 were observed in Δgra7 parasites (Supplementary Fig. 4i) indicating that the high MW bands were not a specific detection of other Toxoplasma proteins (Supplementary Fig. 8c).

To confirm that transmembrane domain-containing GRAs were affected in Δgra45 parasites, we examined the localization of several GRAs after exocytosis of the dense granules based on their membrane association properties: GRA5, is a PVM-integrated protein with its N-terminus faces to the host cytosol;[74,75] and GRA7, associates with IVN and PVM[72,73], and strands extending from PVM into the host cytosol;[76] MAF1, integrates into the PVM with its C-terminus exposed to host mitochondria[77]. We observed that a significantly larger fraction of GRA5 and GRA7 was retained in the PV lumen in Δgra45 parasites compared to wild-type and Δgra45 + GRA45HA parasites (Fig. 4c, d). In Δgra45 parasites, PVM localization of MAF1 was completely abolished at the early infection stage (4 h p.i.) (Fig. 4e) and significantly reduced at 24 h p.i. (Fig. 4f). In addition to transmembrane GRAs, we observed that GRA23, which is an α-helical enriched GRA that is almost exclusively localized to the PVM[78,79], was mislocalized to the PV lumen in Δgra45 parasites (Fig. 4g). GRA23 localization remained normal in Δmyr1 parasites (Supplementary Fig. 4f), which served as a control as it was shown that parasites lacking MYR1 have normal function of PVM-localized GRAs[45]. These results demonstrate that GRA45 plays a critical role in the correct trafficking of PVM-integrated or

-associated GRAs after exocytosis of dense granules. GRA44, as the known GRA45 interaction partner, has no predicted hydrophobic domain but localizes to the PV[49,80]. In contrast to the localization changing from PVM to the PV lumen for above GRAs, GRA44 resided inside the cytosol of Δgra45 parasites instead of being secreted into the PV (Fig. 4h) suggesting that the trafficking of GRA44 through the secretory pathway depends on GRA45, probably via directly interacting with GRA45.

Given that several PVM-localized GRAs and GRA44 appeared to be mislocalized in Δgra45 parasites, we hypothesized that components of the translocon involved in GRA translocation across the PVM could also be mislocalized. To test this hypothesis, we generated Δgra45 parasites in a strain where MYR1 was endogenously tagged with 3xHA epitopes (Supplementary Fig. 4c, j) and determined the solubility of MYR1 in Δgra45 parasites (Supplementary Fig. 8d). Although we did not observe any high MW bands corresponding to MYR1 in Δgra45 parasites, MYR1 was more resistant to detergent extraction and less abundant in the soluble fraction of Δgra45 parasites compared to wild-type parasites (Supplementary Fig. 8e) indicating GRA45 is partially required for the solubility of MYR1. However, it is unclear what happens to the N-terminus of MYR1 in Δgra45 parasites, as MYR1 is cleaved by ASP5 into an N-terminal and C-terminal polypeptide[45,81] and only the C-terminal MYR1 polypeptide was observed for this experiment. Furthermore, MYR2, MYR3, and MYR4 are also involved in GRA effector translocation into the host cell[46,47]. We therefore indirectly assessed if a translocon-mediated phenotype was affected in Δgra45 parasites by determining GRA16 and GRA24 export beyond the PVM. Indeed, GRA16 and GRA24 were no longer exported beyond the PVM in Δgra45 parasites, as observed by their absence from host nuclei, which was similar to what we observed in Δmyr1 parasites (Fig. 4i, j). Overall, these data indicate that GRA45 functions as a potential chaperone protein important for the correct localization of several GRAs to the PVM and the correct trafficking of GRA effectors to the host cell.

**The conserved sHSP signatures are critical for the chaperone-like function of GRA45.** To confirm the chaperone-like function of GRA45, we generated parasites expressing C-terminal HA-tagged GRA45 without its ACD (ΔACD, Fig. 5a and Supplementary Fig. 4k), which is the central structure of sHSPs and the key element for homodimer formation and substrate binding[82]. The parasite strain expressing GRA45 with C-terminal trans-thyretin-like fold deletion (ΔTL) was generated at the same time (Fig. 5a and Supplementary Fig. 4k). To avoid a massive conformational change caused by deletion of a large portion of the protein, we also generated parasites harboring GRA45 with various point mutations including VKV[139–141]/AAA together with

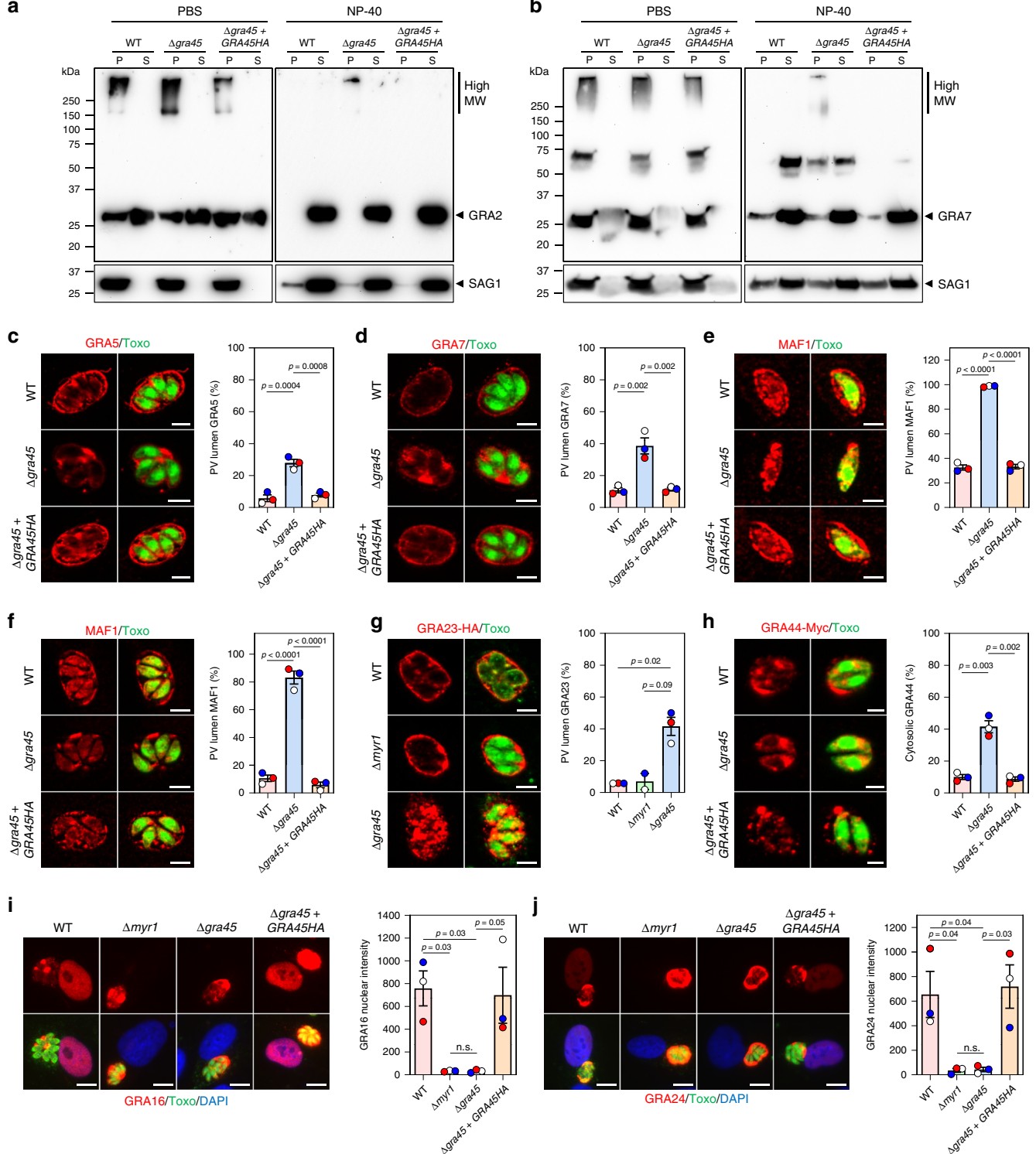

VEV[162–164]/AAA (referred to as VxV/AAA), IDV[205–207]/AAA (referred to as 1st IxV/AAA), IDV[291–293]/AAA (referred to as 2nd IxV/AAA), and IDV[205–207]/AAA together with IDV[291–293]/AAA (referred to as IxV/AAA) (Fig. 5a and Supplementary Fig. 4k). These mutant sites are the conserved I/VxI/V motifs of sHSPs, which direct the formation of sHSP oligomers via binding to hydrophobic pockets of the homodimer[82,83]. We found that in addition to presenting in dense granules of the extracellular parasites all the GRA45 mutants also localized to compartments

peripheral to the parasite nucleus (Fig. 5b), which is reminiscent of the organelles in the early secretory pathway, such as endoplasmic reticulum (ER), Golgi and endosome-like compartment. This change of GRA45 localization in the mutant strains was also observed in the intracellular parasites residing in the vacuole (Fig. 5c), suggesting that a higher level of architecture (e.g., oligomer) is needed for its trafficking to dense granules and secretion. Surprisingly, in these GRA45 mutant parasite strains GRA7 completely colocalized with GRA45 in extracellular parasites

**Fig. 4 In Δ*gra45* parasites other GRAs mislocalized after secretion into the vacuole. a, b** PBS or NP-40-treated pellet (P) and supernatant (S) fraction of extracellular parasite were detected with antibodies against GRA2 (**a**) or GRA7 (**b**). SAG1 was used as the parasite loading control. The image is representative of 4 independent experiments. **c–f** HFFs infected with indicated parasites for 4 h (**e**) or 24 h (**c, d, f**) were stained with antibodies against GRA5 (**c**), GRA7 (**d**), or MAF1 (**e, f**). The images are representative of results from three independent experiments (scale bar = 5 μm). The percentage of vacuoles with only PV-lumen staining was quantified as shown on the right. **g** HFFs infected with indicated parasites transiently expressing GRA23-HA were stained with antibodies against SAG1 (green) and the HA epitope (red). The images are representative of results from three independent experiments for WT and Δ*gra45* parasites with two experiments for Δ*myr1* parasites (scale bar = 5 μm). The percentage of vacuoles with only PV-lumen GRA23 staining was quantified as shown on the right. **h** HFFs infected with indicated parasites transiently expressing GRA44-Myc were stained with antibodies against the Myc epitope. The images are representative of results from three independent experiments (scale bar = 2 μm). The percentage of vacuoles with only cytosolic GRA44 staining was quantified as shown on the right. **i, j** HFFs infected with indicated parasites transiently expressing GRA16-HA-FLAG (**i**) or GRA24-HA-FLAG (**j**) were subjected to the immunofluorescent assay. The images are representative of results from three independent experiments (scale bar = 10 μm). The nuclear intensity of GRA16 (**i**) or GRA24 (**j**) was quantified as shown on the right. Data are displayed as mean ± SEM with independent experiments indicated by the same color dots. Statistical analysis was done by one-way ANOVA with Tukey's multiple comparisons test (**c–f, h–j**) and two-tailed paired *t*-test (**g**) due to different n between Δ*myr1* and Δ*gra45*.

including dense granules as well as the perinuclear compartments (Fig. 5b), suggesting that GRA45 has a chaperone-like function important for the trafficking of GRAs from early secretory organelles to the dense granules. In addition, we noticed that the total amount of GRA7 in Δ*gra45* parasites detected by immunofluorescence assay was significantly less compared to Δ*gra45* + *GRA45HA* parasites (Fig. 5b and Supplementary Fig. 8a) although it still localized to dense granules (long exposure in Supplementary Fig. 8a). This could be due to a large portion of GRA7 remaining insoluble (Fig. 4b) and inaccessible to antibodies in Δ*gra45* parasites. To determine if the localization of GRAs on the PVM was affected in these GRA45 mutant strains, we observed the localization of MAF1 at 4 h (Fig. 5d) and 24 h p.i. (Supplementary Fig. 9b). Like in Δ*gra45* parasites, MAF1 was almost completely retained in the PV lumen of all the mutant strains except for Δ*gra45* + *GRA45_{2nd IxV/AAA}HA* parasites which had ~15% of PVM-localized MAF1 (Supplementary Fig. 9c). These mutations in GRA45 also resulted in the absence of translocation of GRA16 and GRA24 to the host nucleus (Fig. 5e, Supplementary Fig. 9d, e). Since these GRA45 mutant strains have a similar phenotype as Δ*gra45* parasites, we determined the susceptibility of these parasites to IFNγ-induced parasite growth inhibition in murine BMDMs (Fig. 5f). Except for Δ*gra45* + *GRA45_{VxV/AAA}HA* and Δ*gra45* + *GRA45_{2nd IxV/AAA}HA* parasites, all the GRA45 mutant parasites had significantly reduced growth in IFNγ-activated murine BMDMs. Among the mutants, Δ*gra45* + *GRA45_{ΔACD}HA*, Δ*gra45* + *GRA45_{ΔTL}HA*, and Δ*gra45* + *GRA45_{IxV/AAA}HA* parasites had similar level of growth as Δ*gra45* parasites suggesting both the ACD and TL domains as well as both IxV motifs are important for correct trafficking of GRA effector(s) neutralizing the IFNγ responses in murine macrophages. Similar to Δ*gra45* parasites, none of the mutants had a growth defect in naïve murine BMDMs (Supplementary Fig. 5f). Taken together, GRA45 has conserved sHSP functional signatures, which are critical for the trafficking of GRAs (including GRA45 itself) to dense granules and their final destinations.

**GRA45 is important for parasite virulence.** Since the data suggest that GRA45 affects both the correct localization of GRAs to the PVM and the export of GRAs beyond the PVM, it is unclear if the fitness defect of Δ*gra45* parasites in IFNγ-activated BMDMs was due to the defect of PVM-localized GRAs or exported GRAs. However, the translocon components MYR1/2/3/4 did not emerge as hits in our loss-of-function screen in IFNγ-activated BMDMs (Fig. 6a). *GRA45* has a significantly lower IFNγ BMDM vs. Naïve BMDM fitness compared to *MYR1*, *MYR2*, and *MYR3* across the three screens (Fig. 6b). Together with the data that Δ*gra45* + *GRA45_{VxV/AAA}HA* and Δ*gra45* + *GRA45_{2nd IxV/AAA}HA* parasites did not export GRA16 and GRA24 beyond the

PVM (Fig. 5e) but only had minor growth reduction in IFNγ-activated murine BMDMs (Fig. 5f) suggests that GRA export is not involved in resistance to IFNγ-induced parasite growth inhibition. To confirm this, we compared side-by-side the susceptibility of Δ*gra45* and Δ*myr1* parasites to IFNγ-mediated growth inhibition. In murine BMDMs prestimulated with increasing concentrations of IFNγ, we observed that Δ*gra45*, but not Δ*myr1*, parasites were more susceptible than either wild-type or Δ*gra45* + *GRA45HA* parasites to IFNγ-mediated growth inhibition (Fig. 6c). To determine if these in vitro differences also translated to differences in virulence in mice, we intraperitoneally (i.p.) infected mice with 100 tachyzoites of wild-type, Δ*gra45*, Δ*myr1*, Δ*gra45* + *GRA45HA* parasites and monitored parasite virulence. Consistent with the in vitro data, Δ*gra45*, but not Δ*myr1*, parasites were significantly less virulent compared to wild-type or GRA45 complemented parasites (Fig. 6d). These data indicate that MYR-component trafficking and function only partially explain the effects of GRA45 on IFNγ resistance and parasite virulence.

## Discussion
In this study, we performed a genome-wide loss-of-function screen and identified the *Toxoplasma* genes that determine fitness in naïve or IFNγ-activated murine BMDMs. Evading the host immune response at the site of infection and reaching immune-privileged organs is essential for *Toxoplasma* to establish a life-long chronic infection. It is likely that some of the *Toxoplasma* genes that determined fitness at the site of infection were important for resisting the anti-*Toxoplasma* activities in IFNγ-activated cells or determined parasite growth and survival inside macrophages, the cell type preferentially infected by *Toxoplasma* in vivo. We and another group recently determined the in vivo fitness contribution of a subset of *Toxoplasma* genes encoding secretory proteins, mainly including ROPs and GRAs, in the mouse peritoneum and in distant organs[55,84]. Five genes (*GRA22*, *TGGT1_269950*, *GRA38*, *TGGT1_205350*, and *TGGT1_269950*) conferring in vivo fitness were also fitness-conferring genes in IFNγ-activated murine macrophages revealing the in vivo phenotype of these genes is likely due to their importance in maintaining parasite growth in IFNγ-activated peritoneal macrophages.

One of the most significant parasite genes conferring fitness in IFNγ-activated BMDMs was *GRA45*. Unlike previously described GRAs that are involved in parasite virulence, GRA45 is not a parasite effector that directly modulates host cell signaling pathways or inhibits host immunity. Our data indicate that GRA45 is an important chaperone-like protein that determines the correct trafficking of PVM-localized GRA effectors to their final destination, likely via maintaining their native folding status in the

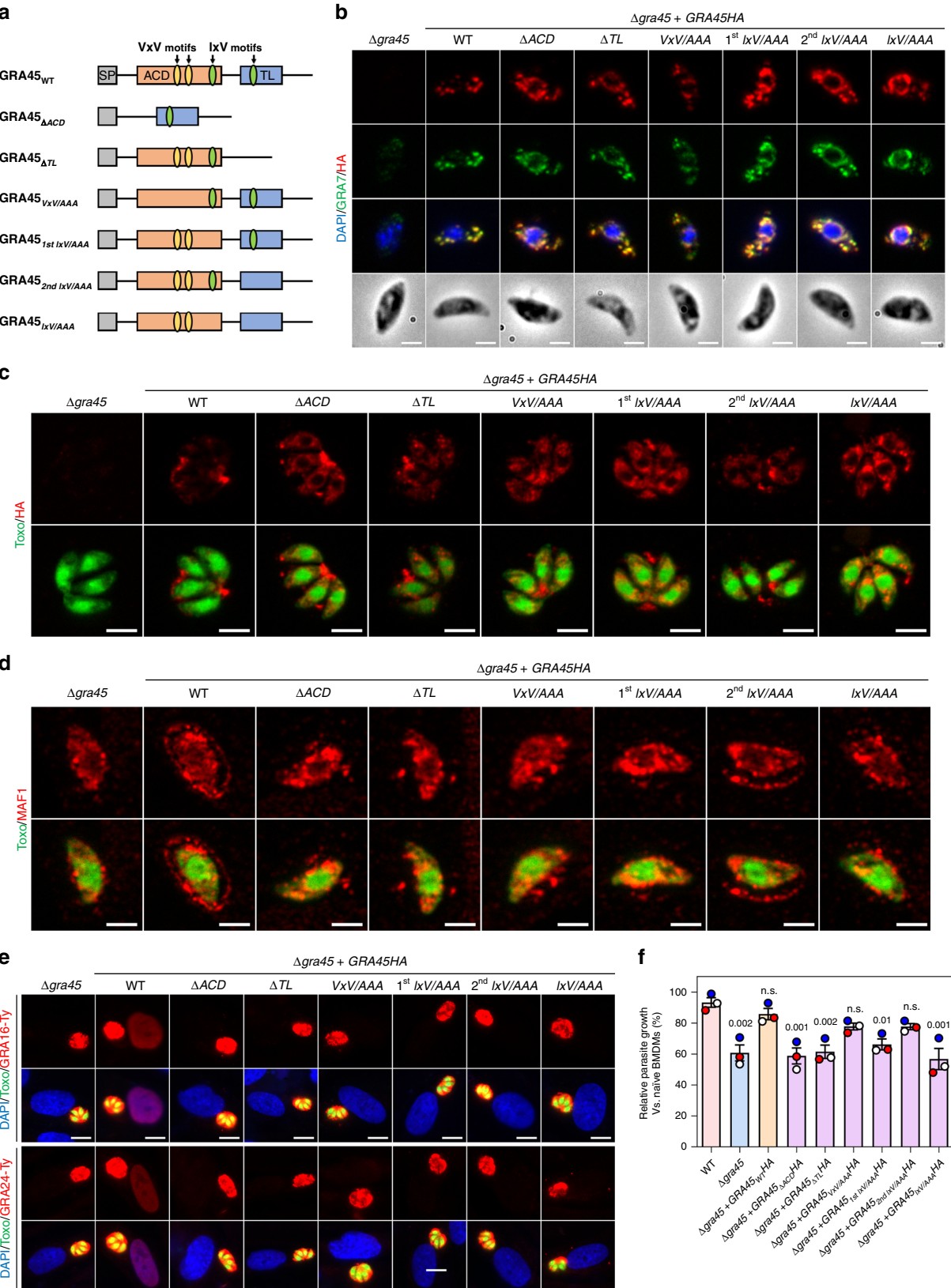

secretory pathway. Among all the characterized ASP5 substrates, GRA45 best phenocopies the mislocalization of GRAs to PVM caused by the deletion of ASP5[48,81]. Due to the absence of a TEXEL motif in some of the PVM-localized GRAs, for example GRA7 and MAF1, it is reasonable to believe that the

mislocalization of GRA7 and MAF1 in Δ*asp5* parasites is due to failed GRA45 processing. Many PVM-integrated or -associated GRAs possess a hydrophobic domain and form high MW complexes while trafficking through the secretory pathway and exist as a mix of soluble and aggregated states before dense granule

**Fig. 5 The α-Crystallin domain (ACD) and I/VxI/V motifs are critical for the chaperone-like function of GRA45. a** Schematic illustration of the constructs used for generation of the parasite expressing full-length GRA45 or GRA45 with indicated mutations. ACD and TL indicate predicted α-Crystallin domain and transthyretin-like fold, respectively. The conserved VxV or IxV motifs are indicated accordingly. **b** Extracellular Δ*gra45* parasites or Δ*gra45* parasites complemented with C-terminal HA-tagged wild-type or indicated mutant version of *GRA45* were fixed with methanol and staining with indicated antibodies. The images are representative of results from two independent experiments (scale bar = 2 µm). **c** HFFs infected with indicated parasite strains for 24 h were fixed and stained with antibodies against HA epitope. The images are representative of results from two independent experiments (scale bar = 5 µm). **d** HFFs infected indicated parasite strains for 4 h were stained with antibodies against MAF1. The images are representative of results from two independent experiments (scale bar = 2 µm). **e** Δ*gra45* or Δ*gra45* complemented with wild-type or indicated mutant version of *GRA45* were transiently transfected with GRA16-Ty (upper) or GRA24-Ty (lower) expressing plasmids and immediately used to infect HFFs and fixed at 24 h p.i. and subjected to the immunofluorescent assay with antibodies against Ty epitope. The images are representative of at least 40 host cells containing a single parasitophorous vacuole with four or more parasites (scale bar = 10 µm). **f** Murine BMDMs prestimulated with 5 ng/mL of IFNγ or left unstimulated for 24 h were infected with indicated parasite strains. Twenty-four hours p.i. parasite growth for each strain was measured by luciferase assay. Parasite growth in IFNγ-activated BMDMs is expressed relative to growth in naïve BMDMs. Data are displayed as mean ± SEM with independent experiments (n = 3) indicated by the same color dots. The significant difference between WT and knockout (or mutant strains) was analyzed with one-way ANOVA with Tukey's multiple comparisons test and the p values are indicated above the columns.

exocytosis[51,85,86]. In bacteria and plants, the sHSPs form dynamic assemblies with the early-unfolding intermediates of their substrates to prevent tight protein aggregate formation upon unfolding stress conditions[82]. The ACD homodimer subunits of sHSP sequester the substrates and preserve them in a complex core structure with a dynamic shell composed of a multitude of diverse oligomerization states driven by the docking of the I/VxI/V motifs to a hydrophobic groove presented in the subunits[82]. Mutations in the I/VxI/V motifs causes dissociation of oligomerized ACD homodimer subunits resulting in loss of the chaperone-like activity[83,87]. Since GRA45 deficient in its conserved sHSP domain/motifs caused transmembrane domain-containing GRAs to partially remain in the early secretory compartments and to mislocalize after secretion, it is likely that GRA45 shields the hydrophobic domain of these GRAs to protect them from uncontrolled aggregation during their trafficking in the secretory pathway. The data also suggest the dimerization/oligomerization of GRA45 is required for its own trafficking to dense granules possibly via an ER/Golgi-resident sorting receptor, which only recognizes GRA45/substrate complexes. Thus, deletion of *GRA45* had pleiotropic effects, which is the most likely explanation for its importance in conferring fitness in IFNγ-activated macrophages of multiple species. In addition to PVM-localized GRAs, GRA16, and GRA24, which are normally secreted beyond the PVM, are no longer secreted in Δ*gra45* parasites. However, it is unlikely that the fitness defect of Δ*gra45* parasites was due to the defect in GRA effector export beyond the PVM as Δ*myr1* parasites were not significantly more susceptible to IFNγ than wild-type parasites. The in vivo fitness of Δ*gra45* parasites also differed from Δ*myr1* parasites, which is consistent with our previous in vivo screen in which *MYR1* had no peritoneum fitness defect[55]. However, studies in the less virulent type II strain showed that individual *MYR1* knockout parasites, but not *MYR1* knockout parasites in a pool of other mutants, lose their virulence in vivo[45,84]. This is most likely due to abolished translocation of the GRA effector *Tg*IST in Δ*myr1* parasites as deletion of *Tg*IST in type II parasites resulted in attenuated virulence in mice[39,40]. However, deletion of *Tg*IST had no effect on parasite susceptibility in IFNγ-prestimulated cells[39,40]. It is therefore likely that the susceptibility of Δ*gra45* parasites is due to the combined effect of mislocalized-PVM proteins. For example, the PVM-localized GRA23 and GRA17 form the nutrient pore in the PVM and GRA23 was mislocalized in Δ*gra45* parasites. *GRA23* also had a ~2-fold stronger fitness defect in Naïve BMDMs vs. HFF control suggesting that *Toxoplasma* might be more reliant on nutrient acquisition via PVM nutrient pores in BMDMs. We cannot exclude the possibility that the majority of IFNγ resistance is dependent on an unknown PVM-localized GRA, of which the

localization to the PVM is mediated by GRA45. Further experiments will be needed to identify such PVM-localized GRA effector(s).

In addition to GRA45, there are other GRAs involved in protein trafficking from parasite plasma membrane to PVM. It was recently shown that phosphorylation of GRAs by WNG1, a closely related family member of WNG2, after exocytosis from the dense granules is important for the correct localization of transmembrane domain-containing GRAs[52]. We have shown that GRA42 and GRA43 also play a role in the correct trafficking of PVM-localized GRAs and that in Δ*gra42* or Δ*gra43* parasites GRA23 and GRA35 are mislocalized[79]. Exactly how WNG1, GRA42, GRA43, and GRA45 function in the GRA trafficking pathway is currently unknown. It is likely that *Toxoplasma* has different PV-lumen-localized chaperones and different transport systems to mediate the trafficking of different classes of membrane proteins (e.g., integral vs. peripheral vs. α-helical). Possibly genes encoding for some of these were hits in our screen. Future more focussed screens and testing additional knockouts of hits from our screen will allow the identification of other *Toxoplasma* genes that mediate the correct trafficking of GRAs to their final destination and understand the precise mechanism by which they influence the localization of GRAs.

In murine cells activated with IFNγ, recruitment of IRGs and GBPs onto the PVM leading to the subsequent vacuole disruption are the predominant mechanisms for controlling *Toxoplasma*[6,9–17]. We found two putative GRAs (TGGT1_263560 and TGGT1_269950) inhibited accumulation of IRGB6 on the PVM, which likely explains their role in parasite fitness in IFNγ-activated murine macrophages. Because the two GRAs were not identified as components of the ROP5/18 complex from previous proteomic studies[27,28], it is unlikely they directly bind to the IRGB6 on the PVM and inhibit its polymerization. A recent study suggested that IRGB6 recruitment is dependent on the recognition of specific phospholipids present on the PVM[88]. It is therefore also possible that the increased IRGB6 loading on the vacuole of Δ*TGGT1_263560* or Δ*TGGT1_269950* parasites is due to a change of PVM lipid composition.

Our RNA-seq data showed that IFNγ modulates cholesterol biosynthesis and many other nutrient pathways in murine macrophages. IFNγ stimulation significantly downregulated the cholesterol biosynthesis pathway in BMDMs as has been shown by others[58]. Thus, it is not surprising that many genes involved in lipid biosynthesis or transport were enriched in IFNγ-activated BMDMs. For example, TGGT1_205350 and TGGT1_242380, members of GNS1/SUR4 family involved in long-chain fatty acid elongation and sphingolipid/ceramide

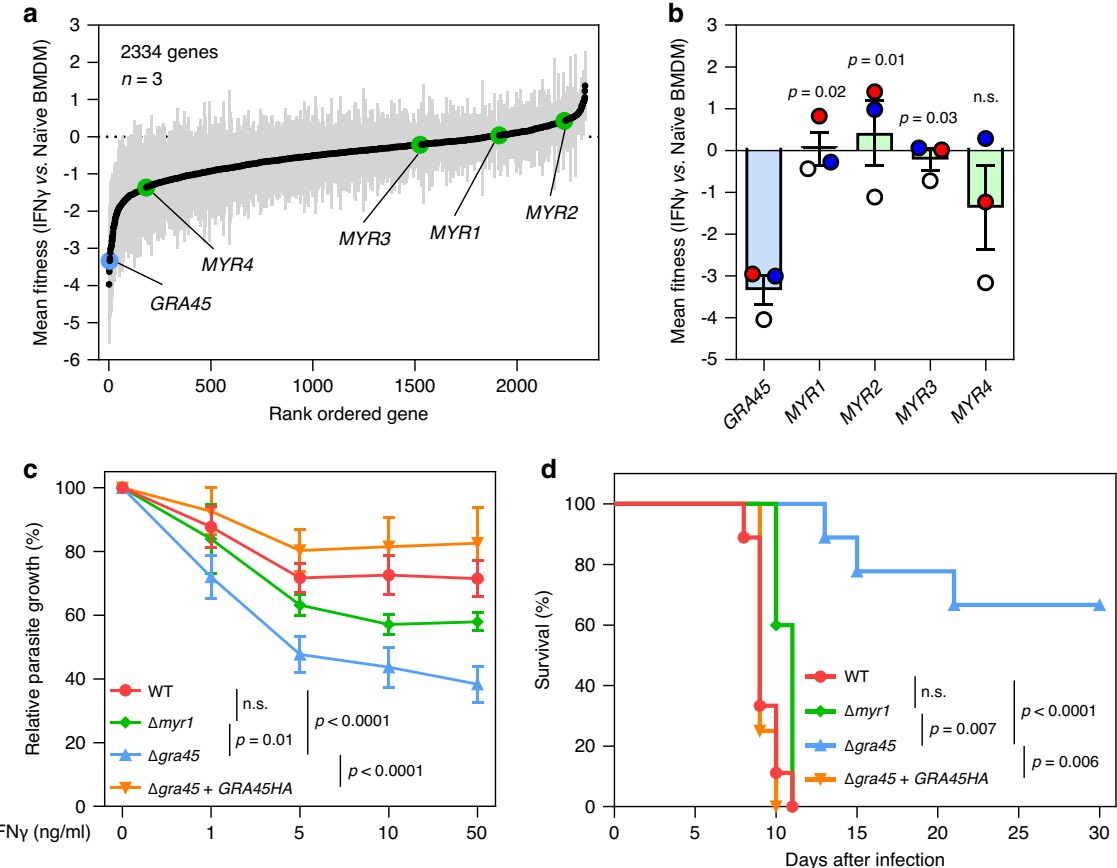

**Fig. 6 Compared to Δ*myr1* parasites, Δ*gra45* parasites are more susceptible to IFNγ and less virulent in mice. a** *Toxoplasma* genes that have at least four sgRNAs present after the 3rd passage in naïve BMDM in the three independent screens are rank-ordered according to their IFNγ vs. Naïve BMDM fitness scores. Data are displayed as average fitness scores (black plots) ± SEM (gray lines). **b** Mean IFNγ vs. Naïve BMDM fitness of *GRA45, MYR1, MYR2, MYR3*, and *MYR4*. Data are displayed as mean ± SEM with independent screens (n = 3) indicated by the same color dots (S1: white; S2: blue; S3: red). The significant difference was analyzed with one-way ANOVA with Tukey's multiple comparisons test. **c** Murine BMDMs prestimulated with or without IFNγ (1, 5, 10, or 50 ng/mL) were infected with luciferase-expressing WT, Δ*myr1*, Δ*gra45*, or Δ*gra45* + GRA45HA parasites (MOI of 0.25) for 24 h. Parasite growth for each strain was measured by luciferase assay and the growth in IFNγ-activated BMDMs is expressed relative to growth in naïve BMDMs. Data are displayed as connecting lines with mean of four independent experiments ± SEM. The significant difference was analyzed with two-way ANOVA with Tukey's multiple comparisons test. **d** CD-1 mice were i.p. infected with 100 tachyzoites of WT, Δ*myr1*, Δ*gra45*, or Δ*gra45* + GRA45HA parasites and survival of mice was monitored for 30 days. Data are displayed as Kaplan–Meier survival curves (n = 9 mice infected with WT or Δ*gra45* parasites, n = 5 mice infected with Δ*myr1* parasites, and n = 4 mice infected with Δ*gra45* + GRA45HA parasites). The significant difference was analyzed with Log-rank (Mantel–Cox) test.

biosynthesis; TGGT1_266640 (Acetyl-coenzyme A synthetase), enzymes involved in fatty acid production and metabolism; GRA38, an orthologue of GRA39 of which the knockout has been shown to accumulate lipid,[63] and ATP-binding cassette transporters ABCG96, which have been shown to mediate cholesterol transport[89]. However, it is well known that *Toxoplasma* is a glutton for host-derived lipids and takes up more than it can safely store in lipid droplets. Because excess lipids, especially free cholesterol, can be toxic, *Toxoplasma* might import more cholesterol and the inability to export excess cholesterol in these cells is likely toxic to the parasite[90]. It is, therefore, possible that *Toxoplasma* needs to modify its acquisition/export or own production of lipids in IFNγ-stimulated BMDMs.

It is worth noticing that there was variation in IFNγ BMDM fitness scores likely resulting in a substantial number of false negatives due to differences in IFNγ concentration and incubation time between the individual screens. For example, deletion of *ROP18* resulted in a fitness defect in IFNγ-activated BMDMs in two out of three screens (IFNγ vs. Naïve BMDM fitness scores of S1, S2, and S3 are −2.03, −0.13, and −1.35, respectively). Given

that S1 and S3 were performed in macrophages activated with 100 ng/mL of IFNγ, parasites deficient in *ROP18* might only be susceptible to growth inhibition induced with a higher amount of IFNγ. *GRA12* had remarkably lower fitness in BMDMs with longer (24 h) IFNγ stimulation (S1 = −3.19 and S2 = −4.15) compared to 4 h of IFNγ incubation (S3 = 0.21) suggesting that the length of IFNγ stimulation probably determined the efficiency of parasite restriction. In addition, our screen likely missed parasite genes that determine fitness in HFFs but that might have an even larger effect on fitness in naïve or IFNγ-activated BMDMs (for example ASP5[48]) because we started our screen after three to four passages in HFFs. At that point, mutants with a fitness defect in HFFs would have already been depleted and therefore, might not have met the threshold to be included in our analysis. The large number of host cells needed to maintain the complexity of the genome-wide library of parasite mutants made it impractical to directly perform the screen in BMDMs. Future focused screens could use the data from our screens as a starting point to identify additional parasite genes that mediate fitness in naïve or IFNγ-activated BMDMs.

## Methods

**Animals**. Six to eight-weeks-old male/female C57BL/6 J mice (Stock No: 000664) and A/J mice (Stock No: 000646) were purchased from The Jackson Laboratory. Six to eight-weeks-old CD-1 female mice were purchased from Charles River Laboratories (Strain Code: 022) and used for in vivo infection experiments. Six-weeks-old female Brown Norway rats (Strain Code: 091) were purchased from Charles River Laboratories. Mice and rats were housed in ventilated cages on corn bedding and provided with water and chow ad libitum. Cages were all on one rack at a housing density of five mice per cage and three rats per cage. The mice and rats were allowed to acclimatize in our vivarium for at least a week undisturbed. The animal room was on a 12-h light/12-h dark cycle, the temperature was maintained at 22–25 °C, and the humidity range was 30–70%. Mice were monitored twice daily by veterinarians, weighed daily, and cage bedding changed every two weeks. Mice and rats were housed under pathogen-specific free conditions at the University of California, Davis animal facility. All animal experiments were performed in strict accordance with the recommendations in the Guide for the Care and Use of Laboratory Animals of the National Institutes of Health and the Animal Welfare Act, approved by the Institutional Animal Care and Use Committee at the University of California, Davis (UC Davis) (assurance number A-3433-01).

**Culture of cells and parasites**. Human foreskin fibroblasts (HFFs, gift from Dr. John C. Boothroyd) were cultured in Dulbecco's modified Eagle's medium (DMEM), 10% fetal bovine serum (FBS), 2 mM L-glutamine, 100 U/mL penicillin/ streptomycin, 10 mg/mL gentamicin. Mouse embryonic fibroblasts (MEFs, gift from Dr. Anthony Sinai) were cultured in DMEM, 10% fetal bovine serum (FBS), 2 mM L-glutamine, 10 mM HEPES, 1x non-essential amino acids, 1 mM sodium pyruvate, 100 U/mL penicillin/streptomycin, 10 μg/mL gentamicin. THP-1 cells (ATCC TIB-202™, gift from Dr. Daniel A. Bachovchin) were cultured in RPMI-1640, 10% fetal bovine serum (FBS), 2 mM L-glutamine, 100 U/mL penicillin/ streptomycin, 10 μg/mL gentamicin. For differentiation, THP-1 cells were stimulated with 100 nM phorbol 12-myristate 13-acetate (PMA, Sigma–Aldrich, Cat#P1585) for 3 days and then rested for 1 day by replacing the differentiation medium with complete medium without PMA before performing experiments. Murine BMDMs were obtained by culturing murine bone marrow cells isolated from the tibia and femur of C57BL/6 J mice or A/J mice in DMEM, 10% fetal bovine serum (FBS), 2 mM L-glutamine, 10 mM HEPES, 1× non-essential amino acids, 1 mM sodium pyruvate, 100 U/mL penicillin/streptomycin, 10 mg/mL gentamicin and 20% L929 conditioned medium for 5–7 days. Rat BMDMs were obtained by culturing rat bone marrow cells isolated from the tibia and femur of Brown Norway rats in DMEM, 20% fetal bovine serum (FBS), 2 mM L-glutamine, 10 mM HEPES, 1× non-essential amino acids, 1 mM sodium pyruvate, 100 U/mL penicillin/streptomycin, 10 mg/mL gentamicin and 30% L929 conditioned medium for 5–7 days.

*Toxoplasma gondii* strains RH-Cas9[53], RH-Cas9Δ*hxgprt*[91], RHΔ*ku80Δhxgprt*[92], RHΔ*ku80::MYR1-3xHA* (kindly provided by Dr. John C. Boothroyd, see ref. [46]), RH-Luc+/Δ*hxgprt*[55], RH-Luc+/Δ*gra22*[55], RH-Luc+/Δ*TGGT1_269950*[55] were routinely passaged in vitro on monolayers of HFFs at 37 °C in 5% $CO_2$.

**Antibodies**. Mouse monoclonal anti-GRA1 (clone TG-17.43) antibodies were purchased from Biotem (Cat#BIO.018.4). Mouse monoclonal anti-GRA2 (clone TG-17.179) and anti-GRA5 (clone TG-17.113) antibodies were purchased from BioVision (Cat#A1298 and Cat#A1299). Mouse monoclonal anti-ROP2/3/4 (clone T34A7) antibodies were described in ref. [93]. Mouse monoclonal anti-SAG1 (clone DG52) antibodies were described in ref. [94]. Rabbit polyclonal anti-GRA7[95], anti-MAF1[96], and anti-SAG1 antibodies were kindly provided by Dr. John C. Boothroyd. Rat anti-HA (3F10) antibodies were obtained from Sigma–Aldrich (Cat#11867431001). Mouse monoclonal anti-Ty (clone BB2) antibodies were described in ref. [53]. Mouse anti-Myc tag (9B11) antibodies were purchased from Cell Signaling Technology (Cat#2276 S). Goat polyclonal anti-IRGA6 antibodies (Cat#sc-11090) and anti-IRGB6 antibodies (Cat#sc-11079) were purchased from Santa Cruz Biotechnology, Inc. Secondary antibodies horseradish peroxidase (HRP)-conjugated Goat anti-Mouse/Rabbit/Rat IgG were purchased from Jackson ImmunoResearch Laboratories Inc. (Cat#111-035-003/#112-035-003/#115-035-003). Goat anti-Mouse IgG (Alexa Fluor 448/594, Cat#A11029/A11032), Goat anti-Rat IgG (Alexa Fluor 594, Cat#A11007), Goat anti-Rabbit (Alexa Fluor 488/594, Cat#A11008/A11037), Donkey anti-Rabbit IgG (Alexa Fluor 488, Cat#A21206), and Donkey anti-Goat IgG (Alexa Fluor 594, Cat#A11058) secondary antibodies were purchased from Thermo Fisher.

**Plasmid construction**. The pU6-Universal plasmid[97] was used for generating the plasmid containing sgRNAs targeting the candidate genes. Briefly, the constructs were generated by annealing oligos containing the sequence of sgRNAs (Supplementary Data 4) which were cloned into the BsaI (New England Biolabs)-digested pU6-Universal plasmid. The pUPRT::DHFR-D (Addgene, Cat#58528) plasmid backbone with PCR-amplification to remove the DHFR cassette was used to generate the construct for making the *GRA45* complementation. The promoter region (~1500 bp upstream to the start codon) and the coding sequence of *GRA45* were amplified and flanked with the HA epitope sequence before the stop codon. The 3′-UTR region (~500 bp) was also amplified and assembled with the other two

fragments using Gibson Assembly (New England Biolabs, Cat#E2611L). For making mutant versions of GRA45 (Fig. 5a), the complementation plasmids containing the wild-type version of GRA45 from above were amplified using the primers listed in Supplementary Data 4 followed by circularizing the PCR products via Q5 Site-Directed Mutagenesis Kit (New England Biolabs, Cat#E0554S). All the GRA45 mutant constructs were confirmed by sequencing. To construct the plasmid for generating the GRA45 endogenously HA-tagged strain, ~1300 bp upstream of the stop codon of GRA45 was amplified by PCR and inserted into pLIC-3xHA-DHFR by ligation-independent cloning[92]. All PCR primers are listed in Supplementary Data 4.

**Generation of parasite strains**. Individual knockout of candidate genes was performed using CRISPR-Cas9[97,98]. To generate the knockout strains for candidate fitness-conferring genes in IFNγ-activated BMDMs, plasmids containing sgRNAs were co-transfected with NotI (New England Biolabs)-linearized pTKO[99], which contains the *HXGPRT* selection cassette and GFP[99], into RH-Luc+/Δ*hxgprt* parasites at a ratio 5:1 of sgRNA to linearized pTKO plasmid. 24 h post-transfection, populations were selected with 50 μg/mL mycophenolic acid (Sigma–Aldrich, Cat#89287) and 50 μg/mL xanthine (Sigma–Aldrich, Cat#X3627) and cloned by limiting dilution (Supplementary Fig. 4a). Individual knockout clones were confirmed by PCR (Supplementary Fig. 4d–f). In addition to generating knockouts in the RH-Luc+/Δ*hxgprt* strain, the RH-Cas9/Δ*hxgprt* strain[91] was used to generate Δ*gra22*, Δ*rom1*, and Δ*gra7* parasites (Supplementary Fig. 4g, i) using the same strategy. To generate the *GRA45* complemented strain (Supplementary Fig. 4b), the RH-Luc+/Δ*gra45* parasites were co-transfected with plasmids containing sgRNAs specifically targeting the *UPRT* locus and SalI (New England Biolabs)-linearized pUPRT::GRA45HA plasmid at a ratio 1:5 of sgRNAs to linearized plasmid. After the first two complete lysis cycles, populations were selected with 10 μM 5-fluoro-2-deoxyuridine (FUDR) (Sigma–Aldrich, Cat#F0503) for another two complete lysis cycles. Individual clones were isolated by limiting dilution and the presence of *GRA45HA* was determined by immunofluorescence assay (IFA) and by PCR to confirm the integration into the *UPRT* locus (Supplementary Fig. 4d). The strains expressing various GRA45 mutations were generated using the same strategy as for the complemented strain construction and confirmed by PCR (Supplementary Fig. 4k). The *GRA45* knockout in the RHΔ*ku80::MYR1-3xHA* strain was generated by co-transfecting plasmids containing sgRNAs targeting the *GRA45* locus along with an amplicon harboring *GRA45* homology regions (60 bp) surrounding a pyrimethamine-resistant (*DHFR**) cassette (Supplementary Fig. 4c). Individual knockout clones were grown in medium supplemented with 3 μM pyrimethamine (Sigma–Aldrich, Cat#46706) followed by limiting dilution and subsequent screening by PCR for correct integration of *DHFR** into the *GRA45* locus (Supplementary Fig. 4j). GRA45 endogenously HA-tagged parasites were made in the RHΔ*ku80Δhxgprt* strain by transfection with the plasmid pLIC-GRA45-3xHA-DHFR. Transfected parasite populations were selected with 3 μM pyrimethamine and cloned by limiting dilution. The presence of *GRA45-3xHA* was determined by immunofluorescence assays.

**Toxoplasma gondii CRISPR-Cas9 mediated genome-wide loss-of-function screens**. At least $5 \times 10^8$ parasites were transfected with a mixture of pU6-DHFR plasmids[53] (100 μg for each $1 \times 10^8$ of parasites) containing 10 different sgRNAs against each of the 8156 *Toxoplasma* genes. A monolayer of HFFs was subsequently infected with the parasites (MOI = 0.5) and grown for 24 h in DMEM containing 1% FBS, 2 mM L-glutamine, 100 U/mL penicillin/streptomycin, and 40 μM chloramphenicol (CAT) (Sigma–Aldrich, Cat#C0378-5). Subsequently, the medium was removed and replaced with DMEM containing 10% FBS, 2 mM L-glutamine, 10 mM HEPES, 1× non-essential amino acids, 1 mM sodium pyruvate, 100 U/mL penicillin/streptomycin, 10 μg/mL gentamicin, 40 μM CAT, 1 μM pyrimethamine, and 10 μg/mL DNase I (New England Biolabs, Cat#M0303S) for three or four passages. To maintain the coverage of the library, at least 10% of the parasites harvested from each lysis were passed to the next round. To perform the screen in murine macrophages, BMDMs isolated from C57BL/6 J mice were stimulated with: 100 ng/mL murine IFNγ (PeproTech, Cat#315-05) for 24 h for screen 1 (S1), 1 ng/mL IFNγ for 24 h for screen 2 (S2), or 100 ng/mL IFNγ for 4 h for screen 3 (S3). Both naïve BMDMs and IFNγ-activated BMDMs were infected with the mutant pool derived from the 3rd or 4th passage in HFFs at an MOI of 0.5 ($1 \times 10^7$ parasites). After 48 h infection (the time the mutant parasites nearly egressed from the host cells), the parasites were harvested, counted and at least 20% of the population were used to infect macrophages that had undergone the same stimulation until the 3rd passage (Fig. 1b). For each passage a pellet of $1 \times 10^7$ parasites were collected and used for genomic DNA extraction and PCR amplification of the sgRNA with a barcoding primer. The sample was sent for Illumina sequencing at the University of California Davis Genomic Center on a NextSeq (Illumina) with single-end reads using primers (P150 and P151) listed in Supplementary Data 4.

sgRNA selection and screen analysis were performed using custom software as previously described[53,100] that will be provided upon request. Statistical analyses were performed in R (www.R-project.org) version 3.6.3 and Excel (Microsoft Office) version 16.0. Illumina sequencing reads were matched against the sequences of the sgRNA library. The number of exact matches was counted and considered as

raw read numbers. The abundance of each sgRNA was normalized to the total number of reads. To do the log$_2$ transfer, sgRNAs that had zero reads were assigned a pseudo-count corresponding to 0.9, which is 90% of the lowest sgRNA read in the sample. Only sgRNA whose abundance was above the 5th percentile in the input (library) were further considered for the analysis. The "fitness" score was calculated as the average log$_2$-fold change of the top five scoring sgRNAs for each given gene, and the mean fitness in the three screens is reported. The Pearson correlation between each sample from different screens was calculated[101]. One-sided Wilcoxon signed-rank test was used to calculate the probability of each gene being fitness conferring by comparing their fitness to the 497 control genes. Cohen's $d$ calculation was used to measure the effect size of fitness scores for each gene. The fitness-conferring genes were defined if they met a significance threshold of $p \leq 0.05$ from one-sided Wilcoxon signed-rank test and Cohen's $d \geq 0.8$ with expression in the murine macrophages (FPKM > 1)[56]. The criteria used to identify the high-confidence genes were described in Table 1.

**RNA-seq.** $3 \times 10^6$ murine BMDMs were seeded overnight in 6-well plates prior to stimulations. For the stimulated samples, IFNγ (100 ng/mL) was added to each well for 4 or 24 h before harvesting the cells for total RNA extractions. The mRNA was purified by polyA-tail enrichment (Dynabeads mRNA Purification Kit, Thermo Fisher, Cat#61006), fragmented into 200–400 basepairs, and reverse transcribed into cDNA before adding Illumina sequencing adapters to each end[102]. Libraries were barcoded, multiplexed into four samples per sequencing lane in the Illumina HiSeq 2000, and sequenced from both ends resulting in 50 bp reads after discarding the barcodes. The RNA-sequencing reads were mapped to the mouse genome (mm10) using Bowtie (version 2.0.2)[103] and Tophat (version 2.0.4)[104] and transcript abundance estimated in cufflinks (version 2.2.1)[105].

**Bioinformatic analysis.** GSEA[57] was used to determine if *Toxoplasma* genes that determine fitness in specific conditions were enriched in the functional annotation. GO enrichment analysis was performed using online tools available at ToxoDB.org. In addition, the genes conferring fitness in specific conditions were analyzed using an in-house database that contained information on GO, protein family domains (Interpro), KEGG enzyme EC numbers, localization to specific organelles, amongst others. Pathways that were enriched are indicated in Supplementary Data 2. For enrichment analysis of BMDM pathways stimulated with IFNγ vs. unstimulated, the GSEA program (version 4.0.3) and Molecular Signatures Database (MSigDB, version 7.1) was used[57,106,107]. Psi-blast was used to find orthologues of proteins under investigation in other species. Alignments of GRA45 with its orthologs were made using PRALINE[108], and the results (Supplementary Fig. 7a) were used as input for HHpred[109] to predict similarity to secondary structures of other proteins. In addition, the SWISS-MODEL server[110] and I-TASSER analysis[70] were used for the homology modeling of protein structures.

**Plaque assay.** Freshly confluent 24-well plates of HFFs or MEFs were infected with 100 parasites for each well and the plates were incubated at 37 °C without disturbing for 5 days. The total number of plaques in HFFs was observed using a Nikon TE2000 inverted microscope. To measure the parasite growth in MEFs, the area of each individual plaque was captured and analyzed with the above microscope equipped with Hamamatsu ORCA-ER digital camera, and NIS Elements Imaging Software, respectively. For each independent experiment, at least 40 plaques from technical duplicate wells were imaged.

**Luciferase assay.** Luciferase assays were performed to determine the fitness of knockout strains in IFNγ-activated macrophages. For murine and rat macrophages, BMDMs in 96-well plates ($1 \times 10^5$ cells/well) were stimulated with 1 ng/mL or 5 ng/mL murine or rat IFNγ (R&D System, Cat#585-IF-100) for 24 h followed by infection with wild-type luciferase-expressing parasites or knockout parasites at two different MOIs (MOI = 0.5 and 1). For human macrophages, PMA-differentiated THP-1 cells in 96-well plates ($1 \times 10^5$ cells/well) were stimulated with 2.5 ng/mL or 5 ng/mL human IFNγ (BIO-RAD, Cat#PHP050) for 24 h followed by parasite infection. A plaque assay was set up in HFFs at the same time to determine the parasite viability of each parasite strain. After 24 h infection, the cells in 96-well plates were lysed using the lysis buffer of Luciferase Assay System (Promega, Cat#E1500) followed by adding luciferin for the measurement of luciferase activity using a single-channel luminometer (Turner BioSystems). Raw luciferase reads (RLU) of unstimulated infected cells was considered as 100 percent and relative parasite growth in IFNγ-activated macrophages was calculated. To make comparisons between wild-type and knockout parasites, "real" MOI was matched from the plaque assay results.

**Growth competition.** The BMDMs were left unstimulated or stimulated with IFNγ (1 ng/mL) for 24 h followed by infection with a 1:1 mixed ratio of GFP-expressing wild-type parasites and GFP-negative knockout parasites (Δgra22 or Δrom1). The mixed parasites were allowed to grow in the BMDMs for three lytic cycles. Plaque assays of the mixed parasites were performed in HFFs before putting into the BMDMs and after the 3rd passage in the BMDMs, and the number of GFP-positive vs. GFP-negative plaques were counted to determine the ratio of wild-type and knockout parasites in naïve and IFNγ-activated BMDMs.

**Immunofluorescent assay.** To determine the localization of GRA45 in extracellular parasites, RHΔku80Δhxgprt::GRA45-3xHA tachyzoites released from syringe-lysed HFFs were loaded onto coverslips and fixed with 4% Paraformaldehyde (PFA) for 20 min followed by permeabilization/blocking with PBS containing 3% (w/v) BSA, 5% (v/v) goat serum and 0.1% Triton X-100. Co-localization was detected by anti-HA antibodies along with antibodies against ROP2/3/4, GRA1, GRA2, GRA5, GRA7, or SAG1. To determine the localization of intracellular GRA45, HFFs grown on glass coverslips were infected with RHΔku80Δhxgprt::GRA45-3xHA parasites for 24 h, fixed with 4% PFA for 20 min, permeabilized/blocked with PBS containing 3% (w/v) BSA, 5% (v/v) goat serum and 0.1% Triton X-100 (or 0.2% Saponin), followed by incubated with antibodies against the HA epitope tag together with antibodies against ROP2/3/4, GRA1, GRA2, GRA5, GRA7, or SAG1 at 4 °C overnight. To check the PVM localization of GRA5 or GRA7, HFFs grown on glass coverslips were infected with wild-type, Δgra45 or Δgra45 + GRA45HA strains in the RH-Luc+/Δhxgprt background for 24 h followed by fixation with 4% PFA, permeabilization with 0.2% Saponin and blocking with PBS containing 3% (w/v) BSA, 5% (v/v) goat serum and 0.1% Saponin. The coverslips were incubated with antibodies against GRA5 or GRA7 for 1 h at room temperature. MAF1 localization was observed in the HFFs infected wild-type, Δgra45 parasites or Δgra45 parasites complemented with wild-type or mutant versions of GRA45. At 4 or 24 h post-infection, the cells were fixed with 4% PFA, permeabilized/blocked with PBS containing 3% (w/v) BSA, 5% (v/v) goat serum and 0.1% Triton X-100 followed by incubation with MAF1 antibodies[96]. To check the PVM localization of GRA23 or export of GRA16 and GRA24, HFFs grown on glass coverslips were infected with wild-type, Δmyr1, Δgra45 parasites or Δgra45 parasites complemented with wild-type or mutant versions of GRA45 transiently expressing GRA23-HA-FLAG, GRA16-HA-FLAG, or GRA16-Ty[50], GRA24-HA-FLAG or GRA24-Ty[50] for 20–24 h followed by fixation, permeabilization and blocking as described above. The coverslips were incubated with anti-SAG1 antibodies and anti-HA antibodies or anti-Ty antibodies at 4 °C overnight. To observe the localization of GRA44, HFFs infected with wild-type, Δgra45 or Δgra45 + GRA45HA strains transiently expressing GRA44-Myc[80] for 20 h were fixed, permeabilized and blocked as described above followed by incubation with anti-Myc antibodies at 4 °C overnight. After incubating with Alexa Fluor 488/594 secondary antibodies and DAPI, the coverslips were mounted with Vecta-Shield mounting oil and the microscopy was performed with NIS-Elements software (Nikon) and a digital camera (CoolSNAP EZ; Roper Scientific) connected to an inverted fluorescence microscope (Eclipse Ti-S; Nikon) and either phase contrast or DIC imaging. To compare the localization of GRAs in various strains, the images were taken at identical exposure times for each channel. To quantify the localization of GRA5, GRA7, and MAF1 at 24 h p.i., at least 100 vacuoles containing four or more parasites were observed. To quantify the localization of GRA23 and GRA44, at least 50 vacuoles containing two or more parasites were observed. To quantify the intensity of GRA16 and GRA24 in host nucleus, at least 30 host cells containing a single PV with 4 or more parasites were observed.

**IRGA6/IRGB6 coating assay.** $1 \times 10^5$ of C57BL/6 J murine BMDMs were plated on coverslips in 24-well plates and rested for 24 h followed by stimulated with 5 ng/mL of murine IFNγ for another 24 h. The stimulated BMDMs were infected with freshly lysed wild-type, Δgra45, Δgra22, ΔTGGT1_263560, or ΔTGGT1_269950 parasites for 1 h followed by washing with PBS for 3–5 times to remove uninvaded parasites, fixation with 4% PFA, permeabilization with 0.2% Saponin and blocking with PBS containing 3% (w/v) BSA, 5% (v/v) goat serum and 0.1% Saponin. The coverslips were incubated with anti-SAG1 antibodies along with antibodies against IRGA6 or IRGB6 for 1 h at room temperature followed by incubation with Alexa Fluor 488/594 secondary antibodies and DAPI for 30 min at room temperature. The coverslips were mounted with Vecta-Shield mounting oil and the microscopy was performed as described above. For each independent experiment, at least 200 vacuoles of each strain were observed and the percentage of IRGA6- or IRGB6-positive vacuoles was quantified.

**Cell fractionation.** Purified extracellular parasites were washed with PBS and resuspended in cold PBS containing 1× protease and phosphatase inhibitors (Thermo Fisher, Cat#78444). Parasites were lysed by six freeze/thaw (F/T) cycles using a dry ice ethanol bath (−70 °C) and a 37 °C bath. 1/3 of the F/T lysate was kept in PBS, 1/3 in 1% NP-40, and 1/3 in 1% Triton X-100 for 30 min at 4 °C under rotation. The PBS, NP-40, and Triton X-100 treated lysates were separated by high-speed spin 30 min at $20,000 \times g$. The PBS supernatant was precipitated with TCA (Trichloroacetic Acid) and the supernatant fraction of NP-40 or Triton X-100 extraction were cold acetone precipitated.

**Immunoblotting.** Samples from different fractions of the extracellular were boiled for 5 min in sample buffer, separated by SDS-PAGE, and transferred to poly-vinylidene difluoride (PVDF) membranes. Membranes were blocked in 5% milk in TBS supplemented with 1% Tween-20 (TBS-T) for 30 min at room temperature and then incubated overnight at 4 °C with primary antibody in the blocking buffer. The GRA2 blot was subsequently incubated with HRP-conjugated anti-mouse secondary antibodies. The GRA7 and SAG1 blots were incubated with HRP-conjugated anti-rabbit secondary antibodies. The HA blot was incubated with

HRP-conjugated anti-rat secondary antibodies. The HRP was detected using an enhanced chemiluminescence (ECL) kit (BIO-RAD, Cat#1705060). Blot pictures were acquired using a FluorChem E system (ProteinSimple). Between each blot, the membranes were stripped by incubation in stripping buffer (Thermo Fisher, Cat#46428) for 20 min and subsequently washed three times with TBS-T. The absence of residual HRP signal on the membrane was tested by using the ECL kit (BIO-RAD, Cat#1705060) and image acquisition. Then, the membrane was blocked prior to incubation with the next primary antibody. The quantification of the bands intensity was done using the Volume Tools of Image Lab 6.1 software (BIO-RAD).

**In vivo infection**. The mice were randomly assigned to experimental groups (the exact number of mice per group is indicated in the figure legends) at the start of the experiment. *Toxoplasma* tachyzoites were harvested from cell culture and released by passage through a 27-gauge needle, followed by a 30-gauge needle. CD-1 mice were intraperitoneally (i.p.) infected with 100 tachyzoites of each strain and parasite viability of the inocula was determined in a plaque assay after infection. The mice were monitored for 30 days p.i., and the number of dead mice per group was observed every individual day.

**Statistical Tests**. All statistical analyses were performed using Prism (GraphPad) version 8.0. All the data are presented as mean ± standard error of mean (SEM), and the exact $n$ values are mentioned in the figure legends. For all the calculations $p < 0.05$ are considered as significant. To compare parasite growth of the knockouts vs. wild-type parasites in IFNγ-activated cells, paired $t$-test was used. For the data with more than two groups with one variable, One-way ANOVA with Tukey's multiple comparisons test was used. For one variable test with two groups, the two-way ANOVA with Tukey's multiple comparisons test was used. Survival experiments were analyzed using the Log-rank (Mantel–Cox) test.

**Reporting summary**. Further information on research design is available in the Nature Research Reporting Summary linked to this article.

## Data availability

The authors declare that all data supporting the findings of this study are available within the article and its Supplementary Information files or are available from the authors upon request. CRISPR screen data including raw sequencing read counts are available in Supplementary Data 1. RNA-sequencing data have been deposited in BioProject database with the accession code PRJNA664106 and are provided in Supplementary Data 3. The MSigDB are available in https://www.gsea-msigdb.org/gsea/msigdb. All unique materials (e.g., the variety of parasite lines described in this manuscript) and custom code used in the analysis of CRISPR screen data are available from the corresponding author (contact: Jeroen P.J. Saeij, jsaeij@ucdavis.edu) upon reasonable request. Source data are provided with this paper.

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

## Acknowledgements

This study was supported by the National Institutes of Health (R01-AI080621) awarded to J.P.J.S. NIH's Director's Early Independence Award (1DP5OD017892), and a Mathers Foundation grant to S.L. We thank W. J. Blakely and G. Arrizabalaga for sharing the GRA44-Myc expressing plasmid; A. Bougdour and M. A. Hakimi for sharing the GRA16-Ty and GRA24-Ty expressing plasmids; and J. C. Boothroyd for sharing the MAF1 antibodies and RHΔ*ku80::MYR1-3xHA* strain. We acknowledge all members of the EupathDB.org team for generating this invaluable resource without which this work would not have been possible.

## Author contributions

J.P.J.S., Y.W., and L.O.S. designed experiments and wrote the manuscript with input from all authors. Y.W. and L.O.S. performed and interpreted most of the experimental work. T.C.P.S. and S.K. helped with the CRISPR screen. M.H. performed the RNA-seq experiments. T.C.P.S. and A.M.F. helped with the generation of knockout parasites and growth competition assays. B.M.M. generated the RH-Cas9Δ*hxgprt* strain. S.L. contributed reagents and analysis tools and provided input on the design of the experiments.

## Competing interests

The authors declare no competing interests.
