## [Peer Review File · Nature Communications]

Reviewers' Comments:

Reviewer #1:

Remarks to the Author:

Wang et al apply CRISPR screening to study Toxoplasma infection of IFN γ stimulated macrophages. Survival against IFN γ defences is key for parasite survival so this is an important condition to address. The authors identify a number of proteins required for parasite survival in naïve and IFN γ treated BMDMs from mice, and test a subset in rat BMDMs and THP-1 cell. They then focus on the investigation of GRA45 in more detail. They propose a chaperone function of GRA45 as a deletion of the gene leads to apparent aggregation of several secreted proteins. I think the GRA45 investigations are an interesting addition to the field. This part of the study is well done- but whether GRA45 acts as a chaperone is speculative- and not supported by data to show that it can interact directly with any of the putative interaction partners.

The screenings are an excellent idea and a well analysed and documented dataset would be a great resource for the community.

But I am not convinced that the analysis is done with sufficient rigor to serve as a resource to the community.

Parts of this may be writing. This is a complicated manuscript to write with the amount of various screening conditions. But there is in general a lack of information on reproducibility between the screens and inclusion of a dataset deemed by the authors to be problematic (S1). It is also surprising that individual experiments are stated to be significant, when higher IFN γ concentration show non-significant results which would be expected to be even more significant. As such it feels that the manuscript would greatly benefit from clearer writing and presentation of the results. There is also a lot of information on the RNAseq data and putative hits that distract from the main story and feel like being thrown in to add data, which does not necessarily fit into the story.

Individual points below:

- 1) I am not sure I understand the logic why S2 is generally better than S1. There are less guides lost in S2, likely because of the milder phenotype ($\sim 15\%$ reduction of growth compared to $\sim 40\%$). But one would expect that the stronger selection gives a more stringent, and more reproducible selection as is seen for example for FUDR selections on UPRT deletion lines.
- 2) It would help to have some numbers on the guideRNAs that went into the screen, and how many were recovered to evaluate how big the bottleneck in the screen really is.
- 3) What proportion of the parasites was maintained at each passage? A small proportion could explain their huge loss of library coverage.
- 4) It appears that in S3, with only a 4h IFN γ treatment the loss of gRNAs is minimized, but we are left a bit in the dark how much is lost based on figure 1C. The colour bar for the HFFP3 in S3 is oversampled, which may obscure the loss of guides in the subsequent passages. A table of how many guides are retained would maybe be more illustrative compared to the heatmaps.
- 5) It is stated that 4 and 24h IFN γ treatment are identical (or near identical). Can the authors provide a correlation plot for that? Figure S1C does look like there are some differences.
- 6) The fitness scores are calculated to the input library. Is that the library that was transfected into HFFs? Why not calculating against the output of the HFFs? This way the authors would remove all HFF fitness-conferring genes from the passage and analysis. It would make the tables much less crowded,

easier to analyse and read.

7) Figure 1d/e: The authors observe a substantial bottleneck in screen S1 and I am not clear whether the fitness scores from that screen are included in the final fitness scores? Because of the substantial loss of guides in S1, I would predict this to affect the overall scores. Are the log₂ fold changes the mean or median of the three different screens? I can't find that information in the manuscript (I may have missed it).

8) S1, the authors argue, is skewed by a severe bottleneck, yet include it in the analysis. If a gene is identified there as being down-regulated as well as in another screen, it is dedicated as a hit. I think this analysis can lead to a lot of random artefacts and if the authors stick to that approach- it would be important to deliver more information. If a gene has a - log₂ fold change in S1 and only one of the other screens and not replicated in the 3rd, this could lead to random inclusion of genes in the candidate list. The fact that the authors use "down-regulation" or p-value adds another layer of random-ness. This I find is a severe weakness of the paper.

9) There are no Venn-diagrams how much overlap they find between the screens, and how much discrepancy.

10) The Gene set enrichment analysis on the Toxo-genes (naïve vs. HFF) are a bit obscure. In most of the enrichments the p-value is not significant. Should the authors not exclude those that are not significantly enriched?

11) The Gene set enrichment for the naïve vs. IFN γ treated are also worrying. Most statistically significant results are from S1, but they are not significant in S3. How do the authors justify the inclusion?

12) Can the authors provide data to show on the scatterplots the 130 and 466 genes they count as having BMDM or IFN γ specific phenotypes? It does not look like that there would be that many outliers.

13) Candidate selection: The authors state that they removed genes from the selection without expression in murine macrophages. How many were in that list? If the number was high- should one be worried?

14) How much overlap between the BMDM experiments and the published in vivo screens is there? If this is only 2 out of the ~466 genes that would be a small number. Can the authors elaborate on that? It may also uncover some interesting results.

15) GRA12 in the table is annotated 2x. One should be called Gra12-like.

16) Figure 3: This is an interesting result but lacks an IFN- control. The authors do not show that Δ Gra22 or Δ Gra45 have any growth defect in RAT BMDMs or THP-1s a priori. I am also surprised to see that the authors chose to use U/mL for the THP-1 experiments and don't show the ng/mL. The result with the higher IFN γ concentration is not significant but the lower concentrations are. This is unexpected as most other experiments show a stronger phenotype. Because of one non-significant and one significant result the authors cannot say this is significant as stated in the text. In conclusion: I don't think the authors can make any statements here without further controls/ experiments to ensure statistical power is sufficient.

17) Have the authors tested all 4 candidates in rat BMDMs and THP-1? They start with 4 KOs and then only talk about the 2 they present. Did the other 2 did not produce any phenotypes?

18) Figure 5c: I am not clear on the statistical comparisons made here. Δ myr1 appears to be reduced halfway between Δ gra45 and WT, there are no error bars for Myr1, or if they are, they are hidden in the graph. Showing p- values here may help.

19) Figure 5d: The myr1 result is interesting given the published essential in vivo phenotype in type-II parasites and should be discussed in the light of the two published in vivo CRISPR screens where

MYR1 does not appear to have a function during peritoneal growth. I am not surprised about this result though as type-I parasites are much more virulent than Type-II. This could be more discussed.

20) The RNAseq is not mentioned in the abstract – please add

21) As you state ≥ 3 in the text please use this in the excel file rather than > 2

22) Line 262 'relatively conserved' – please be more specific

23) Line 281 Please add I-TASSER confidence scores

Reviewer #2:

Remarks to the Author:

T. gondii virulence in mice is dependent on parasitic effectors that are secreted from secretory organelles into the host cell. Some of these proteins localise to the PVM and inactivate IRG/GBP family members. In this respect, ROP5, ROP18 and ROP17 were demonstrated to phosphorylate and inactivate IRGA6 and IRGB6. ROP54 inhibits loading of GBP2. Besides these rhoptry proteins, several dense granule proteins have been identified to contribute to *T. gondii* virulence. Here, the authors initiated a genome-wide-loss-of-function screen to identify novel *T. gondii* type 1 strain proteins, important to circumvent the IFN-induced resistance in murine macrophages. One of the top hits that have been identified was the dense granule protein GRA45. Deletion of *gra45* results in significant attenuation of parasite virulence in vivo and to a somewhat smaller effect in vitro. The authors claim that GRA45 plays a general role in the secretory pathway by preventing aggregation/mislocalisation of other GRA proteins.

General comments:

Although the contribution of GRA45 to parasite virulence in vivo is convincing, its molecular function could not sufficiently be substantiated by the data presented here. It has been described for other GRA proteins (e.g. GRA2 and GRA6) that they contribute to preserve the integrity of the intravacuolar network (IVN) membranes of *T. gondii*. In absence of either *gra2* or *gra6*, the IVN consists of disconnected small vesicles or aggregated material (e.g. Rommerein et al, 2016, Plos One; Travier et al, Int J Parasitol, 2008; Mercier et al, Mol Biol Cell, 2002). In the discussion, the authors themselves say that the absence of *gra42* or *gra43* results in mislocalisation of GRA23 and GRA35. In my opinion, considering GRA45 as a global regulator of secreted GRA effectors (this study) is highly overstated. My major concern is the quality and completeness of data that is directed to demonstrate prevention of other GRA proteins from aggregating/mislocalisation. Besides that, interaction studies are missing to support this conclusion.

Specific comments:

Fig. 2: Why didn't the authors include a *gra46* ko strain?

Why is in b) the difference between 1 and 5 ng IFN in case of the wt strain so tremendous (in all other panels not)?

In panels b) and h) all individual values are not visible. Was h) done only 2x? Dotted lines connecting associated values are difficult to track (especially in g) and h)).

The effect of *gra45* ko is lost with 5 ng IFN. In Fig. 5c the difference between wt and *gra45* ko increases with increasing IFN concentrations (the difference is best at 50 ng IFN). This requires an explanation.

For g) and h) pictures should be provided.

Plaque assay in BMDM and HFF is missing in M&M.

Fig. 3: What is the evidence that TGGT1_263560 is in fact a putative GRA protein?

Suppl. Fig. 3: In panel f) the complemented *gra45* ko strain was done only 3 times (all the other strains have been done 6 times). Please show only the corresponding values (3 in every case) for all the strains here. Alternative: Repeat the experiment with the same number of replicates.

Suppl. Fig. 4:

a) Lane 206/261 (GRA45:GRA44 interaction). Reference is missing.

Why didn't the authors investigate localisation/aggregation of GRA44 in the absence of GRA45? In my view, this would be an informative experiment because an interaction between GRA45 and GRA44 has already been demonstrated.

No indication for structural homology in Suppl. Fig. 4a. Please add.

Amino acid sequence for at least one HSP is missing in Suppl. Fig. 4a. Please add.

b) In case of intracellular parasites, no PVM localization is visible (in case of the proteins for that PVM localization has been described). Why?

The merged pictures have a bad contrast and are therefore difficult to evaluate. Please fix.

Fig. 4:

a) As starting point of the study, the authors used a genome-wide-loss-of-function screen in murine macrophages. At the end, the contribution of GRA45 to virulence was investigated in vivo (mice). Why were all in vitro experiments in between done in human (HFF) cells? In my view, this switch of cell type is difficult to retrace (even if the mechanism/role of GRA45 could be very well conserved in mouse and human cells). Murine fibroblasts are certainly appropriate for all the experiments that were performed in HFF cells.

The intensity of single bands is very inconsistent between single panels:

Why is the GRA2 monomer much more prominent in the NP-40/Tx-100 conditions compared to the PBS condition? If these different intensities reflect different efficiencies of cell lysis, why is the same difference not visible in case of the GRA7 or SAG1 monomers?

The intensity of single bands is very inconsistent within single panels:

Why is the intensity of the GRA2 monomer in the PBS condition very different in wt compared to Δ gra45+GRA45HA? Why is the GRA2 monomer in the wt strain exclusively in the pellet and in case of Δ gra45+GRA45HA exclusively in the supernatant?

Why is the intensity of the high molecular weight complex dramatically different in wt compared to Δ gra45? The explanation here might be that these „aggregates“ contain a lot of different mislocalised or misregulated (GRA) proteins that are unspecifically recognised by the detection antibody. But what is then the exact proportion of the respective protein (GRA2 or GRA7) within this complex? It is impossible to say.

Why is the high molecular weight complex in the NP-40 condition in case of Δ gra45+GRA45HA much more prominent in the supernatant compared to the wt strain (in this case almost nothing is visible in the supernatant)?

If I compare Δ gra45 and Δ gra45+GRA45HA, the intensity of the GRA2 monomer is very different in NP-40 condition but not in the Tx-100 condition. Why? This difference is not visible for the GRA7 monomer. Here, all conditions and strains show the same intensity of the GRA7 monomer. Why?

b) The quality of the Western blots presented here is quite poor. Bands are only partially visible or divided into pieces (e.g. GRA1 monomers). Some signals are heavily distorted (e.g. GRA5 monomers).

Why do the authors strip the membranes between each blot instead of running/staining individual membranes with the individual antibodies? A loading control is missing and should be added. ROP5 (about 60 kDa) could be an option that would not interfere with high molecular weight complexes or monomers. The anti ROP5 antibody was used by the authors before and should therefore be available. Besides these reservations some inconsistencies should be clarified:

The LSS – as the authors state – contains PV membranes and soluble material. Why is GRA7 (a PVM protein) not visible in the LSS (at least not in case of the wt strain)?

For GRA2 and GRA5 monomers, the pattern is exactly the same for all the strains used here. Why? GRA1 WT: Why does the HSS-PBS clearly shows a signal for the monomer but the LSS is completely blank (if I understood correctly the HSS-PBS and HSP-PBS are made from the LSS)? GRA1 is a soluble protein. In that case, why is the GRA1 not present in the LSS anyway? Why does Tx-100 treatment of the HSP-PBS results in a strong band that is absent in the HSP-PBS?

GRA1 Δ gra45: GRA1 is a soluble protein and absence of GRA45 leads to its aggregation. Why is then the LSS monomer so prominent (in comparison to the wt strain)?

Did the authors not properly adjust the volumes? This is a must for these experiments.

The authors should demonstrate localisation/mislocalisation of all the GRA proteins investigated here by immunofluorescence microscopy in addition (like they did for GRA23, GRA16 and GRA24).

Especially for GRA7 monomers, the only difference is the intensity of the phosphorylated form. From this the authors conclude that GRA7 is mislocalised in the absence of GRA45. Is GRA7 exclusively phosphorylated in infected cells (like the authors claim)? In Fig. 4a (using extracellular tachyzoites) one gets the impression that GRA7 also appears at different molecular weights (not clearly visible because the authors show an overexposed membrane). Therefore, the localisation/mislocalisation of GRA7 has to be addressed by an alternative approach (e.g. immunofluorescence).

To confirm the conclusion that GRA45 is a general regulator of other GRA proteins (prevents other GRA proteins from aggregation), GRA45 interaction with different GRA proteins (GRA1, GRA2, GRA5 and GRA7) has to be investigated.

g), i) In my opinion, normalized control data violate all the assumptions of the one-way ANOVA. Can the authors please comment on that issue? Why did they decide to set the control at 100%?

f), h) I don't see any aggregation/mislocalisation of GRA16 (f) or GRA24 (h) in the absence of GRA45 in the pictures provided here.

Fig. 5: The significant in vivo phenotype for Δ gra45 is not reflected by all the in vitro data. Mislocalisation of GRA proteins in absence of GRA45 should be reflected by GBP and IRG (GRA7 does influence Irga6) loading to the PVM. The authors could determine if numbers of vacuoles positive for at least Irga6 change in the absence of GRA45.

Other comments:

Lane 63: The effect of ROP16 and GRA15 was only tested for GBP1 in that study (34).

Lane 65: The effect of ROP54 was only tested for GBP2 in that study (36).

Lane 66: The effect of GRA12 on IRG loading was only tested for Irgb6 in that study (37)

Reviewer #1:

Wang et al apply CRISPR screening to study Toxoplasma infection of IFN γ stimulated macrophages. Survival against IFN γ defences is key for parasite survival so this is an important condition to address. The authors identify a number of proteins required for parasite survival in naïve and IFN γ treated BMDMs from mice, and test a subset in rat BMDMs and THP-1 cell. They then focus on the investigation of GRA45 in more detail. They propose a chaperone function of GRA45 as a deletion of the gene leads to apparent aggregation of several secreted proteins. I think the GRA45 investigations are an interesting addition to the field. This part of the study is well done- but whether GRA45 acts as a chaperone is speculative- and not supported by data to show that it can interact directly with any of the putative interaction partners.

The screenings are an excellent idea and a well analysed and documented dataset would be a great resource for the community. But I am not convinced that the analysis is done with sufficient rigor to serve as a resource to the community. Parts of this may be writing. This is a complicated manuscript to write with the amount of various screening conditions. But there is in general a lack of information on reproducibility between the screens and inclusion of a dataset deemed by the authors to be problematic (S1). It is also surprising that individual experiments are stated to be significant, when higher IFN γ concentration show non-significant results which would be expected to be even more significant. As such it feels that the manuscript would greatly benefit from clearer writing and presentation of the results. There is also a lot of information on the RNAseq data and putative hits that distract from the main story and feel like being thrown in to add data, which does not necessarily fit into the story.

We thank the reviewer for the comments on how to improve our manuscript. Both reviewers have pointed out that the chaperone-like function of GRA45 is speculative. Thus, to support the chaperone-like function of GRA45, we have generated parasite strains containing various GRA45 mutations, including the α -Crystallin domain deletion, and conserved small heat shock protein (sHSP) motifs (I/VxI/V) point mutations, and observed their phenotypes with respect to the localization and translocation of GRAs and IFN γ susceptibility. Our results presented in the new **Fig.5** showed that all these mutant parasites have a similar phenotype as $\Delta gra45$ parasites, such as the mislocalization of MAF1 on the PVM and aborted export of GRA16/24 beyond the PVM, suggesting GRA45 presents a traditional chaperone-like domain (motif) important for the correct trafficking and localization of tested GRAs. In addition, **Fig.5b** and **5c** show that the mutations caused GRA45 to partially remain in the early secretory organelles instead of completely trafficking to dense granules, which resulted in the retention of GRA7 in the same places as GRA45. These results support a chaperone-like function for GRA45 that is important for GRAs trafficking through the secretory pathway and their subsequent correct localization after dense granule exocytosis.

However, we did not perform the experiments to determine the direct interaction between GRA45 and other GRAs due to scientific and technical difficulties: a) sHSPs in other systems usually form dimer/oligomer subunits and the interactions building up the oligomer are rather weak. In addition, the subunits stay dynamic in sHSP/substrate complexes and fluctuate between bound and dissociated states (Mogk et al, *Annu Rev Microbiol*, 2019). In this case, most of the interactions between GRA45 and its substrates could be rather weak and transient. b) some bacterial sHSPs have holdase activity, which has a high and promiscuous binding capacity and interacts with a wide variety of proteins *in vitro* and in cell extracts (Mogk et al, *Annu Rev Microbiol*, 2019). Thus, the *in vitro* interaction experiments have a high chance to have false-positive results. c) Given the special feature of dense granules that hundreds of proteins are packed inside a tiny organelle, immunoprecipitation (IP) of one GRA always pulls-down multiple other GRAs (our own experience but this can also be seen in the GRA BioID data where tagging of one GRA with biotin ligase will label almost all other GRAs). Thus, a better way to determine the specific interaction between GRA45 and its substrates is by quantitative mass-spectrometry comparing the protein abundance of its interaction partners in the immunoprecipitate of wild-type GRA45 vs. mutant GRA45. However, we are unable to do this analysis right now because most proteomics core facilities remain closed except for work on COVID-19. Instead, we have followed the localization of GRA44, a known interaction partner of GRA45 (Coffey et al, *mBio*, 2018), and found that a significantly larger fraction of GRA44 remained inside the cytosol in $\Delta gra45$ parasites indicating that GRA45 is needed for the correct

localization of GRA44 (**Fig.4h**). We believe this result together with the data presented in **Fig.5** are supporting a chaperone-like function of GRA45.

The other reviewer's comments and suggestions mainly focused on data analysis and presentation. We accepted and incorporated the reviewer's suggestions on our screen data analysis and have performed rigorous detailed analyses for assessing the bottleneck of each screen and defining a new set of fitness conferring genes via determining their statistical significance and effect size by comparing with the control genes (see below for details).

The reviewer also stated that it was surprising that individual experiments are stated to be significant, when higher IFN γ concentrations show non-significant results which would be expected to be even more significant. We address this below (point 16). In addition, to make the manuscript easier to understand and better serve as a resource for the community, we have completely rewritten the library screen result section and discussion section, such as simplified the description of screening conditions, changed the flow of data presentation, highlighted the key discoveries, removed the TNF α screen and RNAseq, and reduced the amount of text describing putative hits.

Individual points below:

1) I am not sure I understand the logic why S2 is generally better than S1. There are less guides lost in S2, likely because of the milder phenotype (~15% reduction of growth compared to ~40%). But one would expect that the stronger selection gives a more stringent, and more reproducible selection as is seen for example for FUDR selections on UPRT deletion lines.

The FUDR selections on UPRT deletion lines that the reviewer mentioned is the scenario for a **gain-of-function** screen, in which the *UPRT* gene makes parasites susceptible to FUDR treatment and UPRT-positive parasites die in the FUDR-containing media. In this case, if a CRISPR screen is performed in the presence of FUDR the rare parasite in which the UPRT gene was targeted by sgRNAs would become more abundant after each passage till it eventually takes over the culture. However, our CRISPR/Cas9 screen is a **loss-of-function** screen and the goal of our study was to discover parasite genes that mediate resistance to IFN γ -induced parasite growth inhibition in murine macrophages. Therefore, we have to identify parasites that become less abundant at each passage in the presence of IFN γ . At high concentrations of IFN γ the growth of wild-type parasites (as shown in **Supplementary Fig.1a**) is significantly restricted and therefore at high IFN γ concentrations we would get random loss of parasite mutants if even these mutants are not involved in the phenotype. A strong bottleneck is a double-edged sword in a loss-of-function screen resulting in the random loss of wild-type parasites (even those where the sgRNAs target genes that play no role in IFN γ -resistance) although it will lead to a rapid depletion of parasites mutated in IFN γ -resistance genes. This would result in increased noise. One would therefore predict that the best IFN γ concentration to use is a concentration that doesn't affect the growth of wild-type parasites significantly but would lead to significant growth restriction of parasites in which a gene necessary for survival in IFN γ -activated cells is non-functional.

To visualize the bottleneck caused by different IFN γ stimulation conditions, we have included Lorenz curves in the new **Supplementary Fig.3a** and illustrated the abundance disparity in the sgRNAs from all control genes across different samples. Compared with naïve BMDMs, sgRNA abundance of IFN γ -activated BMDMs decreased in S1 (Gini coefficient (which measures read depth evenness within samples)= 0.90 in IFN γ vs. 0.84 in Naïve), but not S2 (Gini coefficient = 0.81 in both IFN γ and Naïve) and S3 (Gini coefficient = 0.78 in both IFN γ and Naïve BMDMs). Thus, the loss-of-function screens in this study (such as S2 and S3) that have a weaker bottleneck of IFN γ -induced parasite growth restriction are generally better than the one with a bit higher bottleneck (S1) due to a higher chance to have candidate sgRNAs in the top list. It is worth noticing that "fitness" score was determined as the mean log₂ fold change for the top five (but not all) scoring sgRNAs, which minimized the effect of stochastic losses and decreased the error between biological replicates. We have counted the number of genes with certain sgRNAs present and found that control genes with ≥ 5 sgRNAs in the last passage of IFN γ -activated BMDMs were always

significantly enriched (sheet 2 in **Supplementary Data 1**). In addition, instead of analyzing each screen individually we have applied a new statistical analysis (see below) taking into account the three screens together and measured the effect size for each gene to determine the fitness conferring genes by comparing to the control genes. To clarify the potential confusion from the future readers, we have stated that in the methods section from line 650 to 657.

2) It would help to have some numbers on the guideRNAs that went into the screen, and how many were recovered to evaluate how big the bottleneck in the screen really is.

As we mentioned above, we have calculated the Gini coefficient of the abundance disparity in the sgRNAs from all control genes across different samples and added the Lorenz curves correspondingly in the new **Supplementary Fig.3a**. In addition, we have included a sheet containing all the sgRNAs counts for every gene as well as control genes across different samples in the sheet 2 of **Supplementary Data 1**. In this table, we also added the number of genes with certain sgRNA present and determined the enrichment of control genes in the last passages by comparing to the library input.

3) What proportion of the parasites was maintained at each passage? A small proportion could explain their huge loss of library coverage.

To avoid random loss caused by taking a small proportion of the harvested parasites, at least 10% of harvested parasites were passed to the next infection during pyrimethamine selection in HFFs while at least 20% were passed into the murine BMDMs. The number of passaged parasites is sufficient for at least 100x coverage of the library. We have added the information to the method section from line 631 to 632 and line 637 to 638.

4) It appears that in S3, with only a 4h IFN γ treatment the loss of gRNAs is minimized, but we are left a bit in the dark how much is lost based on figure 1C. The colour bar for the HFFP3 in S3 is oversampled, which may obscure the loss of guides in the subsequent passages. A table of how many guides are retained would maybe be more illustrative compared to the heatmaps.

We have added a table containing the total number of sgRNAs from all genes or control genes across different passages into the new sheet 2 of **Supplementary Data 1**. As above, we have also included the number of genes that are represented by a certain number of sgRNAs in the table. The sgRNA abundance of control genes in different passages was illustrated in the new **Supplementary Fig. 3a** with Lorenz curves instead of the heatmaps.

5) It is stated that 4 and 24h IFN γ treatment are identical (or near identical). Can the authors provide a correlation plot for that? Figure S1C does look like there are some differences.

We have provided the correlation plot in the new **Supplementary Fig. 2b**. The transcriptional profile of the IFN γ -regulated genes (4 fold up- or down-regulation) in the murine BMDMs stimulated with IFN γ for 24 h is highly correlated with the one after 4 h of IFN γ stimulation ($r = 0.85$ for C57BL/6J BMDMs and $r = 0.83$ for A/J BMDMs) whereas the correlation with BMDMs stimulated for 24 h with TNF α (we have removed the RNAseq data of TNF α -stimulated BMDMs in the manuscript) was low ($r = 0.45$ for C57BL/6J BMDMs and $r = 0.46$ for A/J BMDMs). Despite the high correlation between 4 and 24 h of IFN γ stimulation, a portion of the upregulated genes has minor induction with 4 h of IFN γ stimulation while ~65% of downregulated genes have less than 2 fold down-regulation. Therefore, we stated that “IFN γ stimulation for 4 and 24 h induced the expression of **similar** gene sets” in the initial submission. To clarify this further, we have modified the text as “IFN γ (100 ng/mL) stimulation of murine BMDMs for 4 h induced a similar but slightly lower expression of gene sets that had ≥ 2 fold upregulation in IFN γ stimulation for 24 h” from line 147 to 149.

6) The fitness scores are calculated to the input library. Is that the library that was transfected into HFFs? Why not calculating against the output of the HFFs? This way the authors would remove all HFF fitness-conferring genes from the passage and analysis. It would make the tables much less crowded, easier to analyse and read.

Yes, the input is the pool of linearized library plasmids that was used for the transfection into the parasites. Before transfecting into the parasites, we have always taken a portion of linearized library plasmids for further PCR amplification and Illumina sequencing at the same time when we processed the parasite samples.

It is worth noticing that the mathematical values of fitness score between our current calculation and the reviewer-suggested calculation are mostly identical regardless of the input as shown in this formula: $\text{Log}_2(\text{IFN}\gamma/\text{Input}) - \text{Log}_2(\text{Naïve}/\text{Input}) = \text{Log}_2[(\text{IFN}\gamma/\text{Input})/(\text{Naïve}/\text{Input})] = \text{Log}_2(\text{IFN}\gamma/\text{Naïve})$. In our previous *in vitro* studies, the fitness scores were always calculated by comparing to the input library (Sidik et al, *Cell*, 2016; Sidik et al, *Nat Protoc*, 2018; Waldman et al, *Cell*, 2020). We believe the consistent description of fitness score calculation will benefit the community as our colleagues could easily understand the calculation from this study and compare the fitness of a gene of interest across different studies generated from our groups and determine the potential function of the gene in specific conditions.

We agree with the reviewer that Table 1 is a bit crowded. Thus, we have simplified the text of fitness score calculation from line 124 to 127 and removed/changed some columns from Table.1.

7) Figure 1d/e: The authors observe a substantial bottleneck in screen S1 and I am not clear whether the fitness scores from that screen are included in the final fitness scores? Because of the substantial loss of guides in S1, I would predict this to affect the overall scores. Are the log₂ fold changes the mean or median of the three different screens? I can't find that information in the manuscript (I may have missed it).

We have first calculated the mean log₂ fold change for the top five scoring sgRNAs from each individual screen as the “fitness” score and then reported the mean fitness of a given gene from three screens as the final fitness, which was used for generating **Fig.1d** and **1e** as well as presented in **Table 1** and **Supplementary Data 1**. We have now changed the axis titles to “Mean fitness” in the new **Fig.1c** and **1b**, and added the information into the methods section from line 650 to 652.

We agree with the reviewer that the S1 generally has a lower IFN γ vs. Naïve BMDM fitness score for many genes compared to the other two screens resulting in a slightly lower overall fitness score. To overcome the problem raised by the reviewer we have analyzed the probability of conferring fitness for each gene by comparing the distribution of its fitness scores across the three screens to the average fitness score of all control genes using the Wilcoxon signed-rank test. In addition, Cohen's *d* calculation was used to determine the effect size of fitness scores for each gene. Genes expressed in the murine macrophages (Melo et al, *PLoS Pathog*, 2013) were considered fitness-conferring if they met a significance threshold of *p*-value < 0.05 (Wilcoxon signed-rank test) and Cohen's *d* \geq 0.8 (stands for large effects). By doing so, we have defined 160 genes as the IFN γ -conferring fitness gene and all of our confirmed hits are presented in the list. In addition, we have applied this way of analysis for comparing Naïve BMDM with HFF_P6/P8 (now called HFF control) and defined 193 genes that are required for the parasite growth in naïve murine macrophages. We have now modified the figures and text based on the new analysis accordingly. Note that if one would like to have even more stringent criteria in **Supplementary Data 1** the fitness scores for each screen are indicated. By requiring that these 160 fitness-conferring genes have a 2-fold fitness defect in each of the 3 screens 47 genes would remain. Although the exact number of fitness conferring genes in Naïve or IFN γ -activated BMDM is an approximation it is worth pointing out that of the 7 high-confidence candidates we have tested all had a phenotype in IFN γ -activated macrophages.

8) S1, the authors argue, is skewed by a severe bottleneck, yet include it in the analysis. If a gene is identified there as being down-regulated as well as in another screen, it is dedicated as a hit. I think this analysis can lead to a lot of random artifacts and if the authors stick to that approach- it would be important to deliver more information. If a gene has a – log₂ fold change in S1 and only one of the other screens and

not replicated in the 3rd, this could lead to random inclusion of genes in the candidate list. The fact that the authors use “down-regulation” or p -value adds another layer of random-ness. This I find is a severe weakness of the paper.

We thank the reviewer for pointing this out and we agree with the reviewer that the original analysis used in the initial submission defining candidates by either down-regulation (negative fitness score) or p -value from MAGeCK analysis might not have been sufficiently stringent. Thus, to make the screen data more convincing we decided to remove the p -value from MAGeCK analysis because the algorithm was only performed for each individual screen, and the average p -value from three screens might be susceptible to outliers. As we answered in the last comment, we carried out a more reliable statistical analysis for determining the fitness-conferring genes to overcome the randomness of candidate gene selection. We now present a list of genes with a significance threshold of p -value < 0.05 (Wilcoxon signed-rank test) and Cohen's $d \geq 0.8$. The new lists of candidate genes were used for further analysis and generating the **Fig.1c** and **1d**. Despite the fact that we have changed the way of analysis, the final top candidate genes presented in **Table.1** are mostly identical revealing that these genes (with seven we have already confirmed) are indeed the high-confidence hits from the study. We would like to again point out that the fitness scores for each screen for each gene are presented and therefore someone can set their cut-offs as stringent as they want or if they wish could only focus on the data from S2 and S3.

9) There are no Venn-diagrams how much overlap they find between the screens, and how much discrepancy.

With the current way of analysis, we have determined the fitness-conferring genes by analyzing the three screens together instead of performing analysis and defining the hits from each individual screen. Thus, we could not provide the Venn-diagrams for the overlap between screens. However, we believe the new p -value from the Wilcoxon signed-rank test together with Cohen's d for the effect size already represents the consistency for a given gene across the three individual screens.

10) The Gene set enrichment analysis on the Toxo-genes (naïve vs. HFF) are a bit obscure. In most of the enrichments the p -value is not significant. Should the authors not exclude those that are not significantly enriched?

We agree with this reviewer that the p -values were not significant, and probably the main reason for having this problem was that we analyzed the three screens individually. Since a new set of candidate genes was defined, we have performed both the gene set enrichment analysis (GSEA) with our in-house database and gene ontology (GO) enrichment analysis through ToxoDB.org. Because the enrichment gene sets were similar between both analyses, we therefore presented the results from GO analysis as the new **Supplementary Data 2**.

11) The Gene set enrichment for the naïve vs. IFN γ treated are also worrying. Most statistically significant results are from S1, but they are not significant in S3. How do the authors justify the inclusion?

As we have answered above, we have redone the enrichment analysis with the newly defined IFN γ candidate genes and presented the results as the new **Supplementary Data 2**.

12) Can the authors provide data to show on the scatterplots the 130 and 466 genes they count as having BMDM or IFN γ specific phenotypes? It does not look like that there would be that many outliers.

Since new ways of data analysis and candidate selection were applied, we have provided the distribution of 193 genes with a naïve macrophage fitness defect and 160 genes with IFN γ fitness defect as additional scatterplots in the new **Supplementary Fig. 3b** and **3c** illustrated by mean fitness as X-axis and $-\log_{10}(p$ -value) as Y-axis. Note that the p -value in this graph was calculated from the Wilcoxon signed-rank test as

described above and is different from the p -values in our first manuscript which came from the MAGeCK algorithm. We agree with the reviewer that it looks like not many outliers appeared in the figures from initial submission because the graphs are too compact with many dots overlapping so that it is hard to visualize the genes with less but significantly lower fitness scores in the graph. Thus, we have only highlighted the high-confidence genes listed in **Table.1** in the new **Fig.1c** and **1d** as we mainly emphasized and discussed the potential function of these genes in the manuscript.

13) Candidate selection: The authors state that they removed genes from the selection without expression in murine macrophages. How many were in that list? If the number was high- should one be worried?

Without applying this criterion only one gene (TGGT1_228420) with FPKM < 1 presented in the list of high-confidence candidates for naïve BMDM fitness whereas all the IFN γ high-confidence genes have FPKM > 1. To make the information available for the reviewer, we have added a column with the FPKM value in the murine macrophages from our previous study (Melo et al, *PLoS Pathog*, 2013) in the new Supplementary Data 1.

14) How much overlap between the BMDM experiments and the published *in vivo* screens is there? If this is only 2 out of the ~466 genes that would be a small number. Can the authors elaborate on that? It may also uncover some interesting results.

We have double-checked the overlap of our newly defined fitness-conferring genes (193 with macrophages fitness defect and 160 with IFN γ fitness defect) with our previously published *in vivo* screen. There are four genes that overlap, *GRA22*, *TGGT1_269950*, *GRA38*, and *TGGT1_205350*, of which we have confirmed *GRA22* and *TGGT1_269950* in this study. We also checked the overlap with another *in vivo* CRISPR screen (Young et al, *Nat Commun*, 2019) and found that *TGGT1_299780* from that study is in our IFN γ candidate list. Thus, we have mentioned *GRA38*, *TGGT1_205350*, and *TGGT1_299780* in the discussion from line 413 to 414. The main reason that the overlap between our study and published *in vivo* screens is small is that the *in vivo* screens only tested a subset of genes encoding known and putative ROPs and GRAs whereas our screen evaluated genes on a whole-genome scale. In addition, we have only assessed the genes that are important for parasite growth in murine macrophages. As several studies have previously shown, multiple cell types are involved *in vivo* pathogenesis of *Toxoplasma* although macrophages are the predominant cells of the parasite-infected *in vivo*.

15) GRA12 in the table is annotated 2x. One should be called Gra12-like.

We thank the reviewer for pointing this out, we have changed the “TGGT1_275850” to “GRA12-like”.

16) Figure 3: This is an interesting result but lacks an IFN- control. The authors do not show that Gra22 or Gra45 have any growth defect in RAT BMDMs or THP-1s a priori. I am also surprised to see that the authors chose to use U/mL for the THP-1 experiments and don't show the ng/mL. The result with the higher IFN γ concentration is not significant but the lower concentrations are. This is unexpected as most other experiments show a stronger phenotype. Because of one non-significant and one significant result the authors cannot say this is significant as stated in the text. In conclusion: I don't think the authors can make any statements here without further controls/ experiments to ensure statistical power is sufficient.

Both $\Delta gra45$ and $\Delta gra22$ parasites have no growth defect in rat BMDMs and THP-1 macrophages. We have now added the data along with the growth of $\Delta TGGT1_263560$ and $\Delta TGGT1_269950$ parasites in both cells as the new **Supplementary Fig. 6**. In addition, we have converted the “U/mL” for human IFN γ to “ng/mL” based on the manufacturer's instruction.

We disagree with the reviewer that it is unexpected that results with a high IFN γ concentration are not significant. We believe part of this confusion is due to a lack of statistical power when using high IFN γ concentration. One can imagine that with increasing IFN γ concentrations the wild-type parasite's growth is increasingly inhibited. When wild-type growth is significantly inhibited (e.g., 70%) the difference between

wild-type and mutant can mathematically only be maximally 30% (if the mutant is inhibited 100%). However, at lower IFN γ concentrations when wild-type growth is only minimally inhibited the difference with the mutant can be maximal. This is the reason we often perform these growth inhibitions at different IFN γ concentrations. Since the reviewer was concerned about the control for this data, we performed a new set of experiments in THP-1 macrophages testing various knockouts at the same time by using another concentration of human IFN γ (5 ng/mL), which caused ~35% of growth inhibition in wild-type parasites. Compared to both wild-type and $\Delta gra45 + GRA45HA$ parasites, $\Delta gra45$ parasites, but not $\Delta TGGT1_263560$ and $\Delta TGGT1_269950$ parasites, had significantly reduced growth in IFN γ -activated THP-1 macrophages (as shown in the below figure). The data are included in the new Fig. 3e, 3g, and 3h. However, to not disturb the flow, we did not present the data from $\Delta gra45 + GRA45HA$ parasites in the manuscript.

17) Have the authors tested all 4 candidates in rat BMDMs and THP-1? They start with 4 KOs and then only talk about the 2 they present. Did the other 2 did not produce any phenotypes?

We originally did not test the other two putative GRAs (TGGT1_263560 and TGGT1_269950) in the initial submission. However, to give the reviewer an answer to the question, we have tested the growth of $\Delta TGGT1_263560$ and $\Delta TGGT1_269950$ parasites in IFN γ -activated rat BMDMs as well as THP-1 macrophages. Both strains showed a similar level of growth compared to the WT in these two cells and the data are included in the new Fig. 3b, 3g, and 3h.

18) Figure 5c: I am not clear on the statistical comparisons made here. *myr1* appears to be reduced halfway between *gra45* and WT, there are no error bars for Myr1, or if they are, they are hidden in the graph. Showing p-values here may help.

Unfortunately in the original submission the error bars for $\Delta myr1$ parasites were too short to see and were hidden by the dots. To show the error bars, we have reduced the size of the dots displayed in the graph. For the statistical test carried out in this figure, the two-way ANOVA with Tukey's multiple comparisons was applied for comparing the whole curve (overall relative parasite growth in BMDMs pre-stimulated with different concentration of IFN γ) of each strain with the other strains instead of comparing the relative parasite growth at one specific concentration of IFN γ . In this analysis, the *p*-value of $\Delta myr1$ vs. WT and $\Delta gra45$ vs. $\Delta myr1$ is 0.15 and 0.01, respectively. We have performed an additional statistical test to compare the relative growth of $\Delta myr1$ parasites with WT parasites at each particular concentration of IFN γ using the two-way ANOVA. The *p*-values are 0.97, 0.76, 0.28, and 0.40 for 1 ng/mL, 5 ng/mL, 10 ng/mL, and 50 ng/mL of IFN γ , respectively. Thus, MYR1 is not a fitness-conferring gene for parasite growth in IFN γ -activated murine BMDMs.

19) Figure 5d: The *myr1* result is interesting given the published essential *in vivo* phenotype in type-II parasites and should be discussed in the light of the two published *in vivo* CRISPR screens where MYR1 does not appear to have a function during peritoneal growth. I am not surprised about this result though as type-I parasites are much more virulent than Type-II. This could be more discussed.

We thank the reviewer for this suggestion and have discussed the *in vivo* phenotype of MYR1 from line 446 to 450.

20) The RNAseq is not mentioned in the abstract – please add

Given the 150 words limit in the abstract plus that the RNAseq is not the primary data for the study, we unfortunately can not mention the RNAseq in the abstract.

21) As you state ≥ 3 in the text please use this in the excel file rather than >2

We have changed to “ \geq ” throughout the text and tables including Supplementary Data.

22) Line 262 ‘relatively conserved’ – please be more specific

We have indicated the percentage of amino acid identities for GRA45, GRA44, and WNG2 with their orthologues from line 245 and 246.

23) Line 281 Please add I-TASSER confidence scores

We have included the TM-score of the top 10 models with GRA45 structural homology predicted by I-TASSER from line 262.

Reviewer #2:

T. gondii virulence in mice is dependent on parasitic effectors that are secreted from secretory organelles into the host cell. Some of these proteins localise to the PVM and inactivate IRG/GBP family members. In this respect, ROP5, ROP18 and ROP17 were demonstrated to phosphorylate and inactivate IRGA6 and IRGB6. ROP54 inhibits loading of GBP2. Besides these rhopty proteins, several dense granule proteins have been identified to contribute to *T. gondii* virulence. Here, the authors initiated a genome-wide-loss-of-function screen to identify novel *T. gondii* type 1 strain proteins, important to circumvent the IFN-induced resistance in murine macrophages. One of the top hits that have been identified was the dense granule protein GRA45. Deletion of *gra45* results in significant attenuation of parasite virulence in vivo and to a somewhat smaller effect in vitro. The authors claim that GRA45 plays a general role in the secretory pathway by preventing aggregation/mislocalisation of other GRA proteins.

General comments:

Although the contribution of GRA45 to parasite virulence in vivo is convincing, its molecular function could not sufficiently be substantiated by the data presented here. It has been described for other GRA proteins (e.g. GRA2 and GRA6) that they contribute to preserve the integrity of the intravacuolar network (IVN) membranes of *T. gondii*. In absence of either *gra2* or *gra6*, the IVN consists of disconnected small vesicles or aggregated material (e.g. Rommerein et al, 2016, Plos One; Travier et al, Int J Parasitol, 2008; Mercier et al, Mol Biol Cell, 2002). In the discussion, the authors themselves say that the absence of *gra42* or *gra43* results in mislocalisation of GRA23 and GRA35. In my opinion, considering GRA45 as a global regulator of secreted GRA effectors (this study) is highly overstated.

My major concern is the quality and completeness of data that is directed to demonstrate prevention of other GRA proteins from aggregating/mislocalisation. Besides that, interaction studies are missing to support this conclusion.

We thank the reviewer for the comments on how to improve our manuscript. We agree with this reviewer that considering GRA45 as a global regulator of secreted GRA effectors in this study was overstated and not sufficiently backed up by experimental evidence. In the current version of the manuscript, our data demonstrate that GRA45 influences the correct localization of several PVM-associated GRAs by directly observing the PVM localization (e.g. GRA5/GRA7/GRA23/GRA44/MAF1) or quantifying the downstream phenotypes of PVM-localized GRAs (e.g MYR complex-mediated GRA16/24 nuclear translocation). We have removed the text that "GRA45 is a global regulator of secreted GRAs" and rephrased the description of GRA45 in the result and discussion sections toward its role in the trafficking/localization of other GRAs.

As we responded to reviewer #1, to support the chaperone-like function of GRA45, we have generated parasite strains containing various GRA45 mutations, including the α -Crystallin domain deletion, and conserved small heat shock protein (sHSP) motifs (I/VxI/V) point mutations, and observed their phenotypes with respect to the localization and translocation of GRAs and IFN γ susceptibility. Our results presented in the new **Fig.5** showed that all these mutant parasites have a similar phenotype as Δ *gra45* parasites, such as the mislocalization of MAF1 on the PVM and aborted export of GRA16/24 beyond the PVM, suggesting GRA45 presents a traditional chaperone-like domain (motif) important for the correct trafficking and localization of tested GRAs. In addition, **Fig.5b** and **5c** show that the mutations caused GRA45 to partially remain in the early secretory organelles instead of completely trafficking to dense granules, which resulted in the retention of GRA7 in the same places as GRA45. These results clearly indicated that GRA45 has a chaperone-like function, which is important for GRAs trafficking through the secretory pathway and their subsequent correct localization after dense granule exocytosis. However, we did not perform the experiments that directly address the interaction between GRA45 and other GRAs due to scientific and technical difficulties: a) sHSPs in other systems usually form dimer/oligomer subunits and the interactions building up the oligomer are rather weak. In addition, the subunits stay dynamic in sHSP/substrate complexes and fluctuate between bound and dissociated states (Mogk et al, *Annu Rev Microbiol*, 2019). In this case, most of the interactions between GRA45 and its substrates could be rather weak and transient. b) some bacterial sHSPs have holdase activity, which has a high and promiscuous

binding capacity and interacts with a wide variety of proteins *in vitro* and in cell extracts (Mogk et al, *Annu Rev Microbiol*, 2019). Thus, the *in vitro* interaction experiments have a high chance to have false-positive results. c) Given the special feature of dense granules that hundreds of proteins are packed inside a tiny organelle, immunoprecipitation (IP) of one GRA always pulls-down multiple other GRAs (our own experience but this can also be seen in the GRA BioID data where tagging of one GRA with biotin ligase will label almost all other GRAs). Thus, a better way to determine the specific interaction between GRA45 and its substrates is by quantitative mass-spectrometry comparing the protein abundance of its interaction partners in the immunoprecipitate of wild-type GRA45 vs. mutant GRA45. However, we are unable to do this analysis right now because most proteomics core facilities remain closed except for work on COVID-19. Instead, we have followed the localization of GRA44, a known interaction partner of GRA45 (Coffey et al, *mBio*, 2018), and found that a significantly larger fraction of GRA44 remained inside the cytosol in $\Delta gra45$ parasites indicating that GRA45 is needed for the correct localization of GRA44 (**Fig.4h**). We believe this result together with the data presented in **Fig.5** are supporting a chaperone-like function of GRA45.

Specific comments:

Fig. 2:

Why didn't the authors include a *gra46* ko strain?

We have not generated the GRA46 knockout strain yet. However, we believe seven confirmed candidate genes are sufficient to verify the credibility of our screen.

Why is in b) the difference between 1 and 5 ng IFN in case of the wt strain so tremendous (in all other panels not)?

We understand the reviewer's concern on the variation of IFN γ -induced WT parasite growth restriction in **Fig. 2a to 2f**. The experiments were performed in the primary murine BMDMs isolated from two different batches and the results obtained from the different batches of cells were relatively variable due to the variability that exists between donors and preparations. Thus, the susceptibility of BMDMs in response to IFN γ stimulation is slightly different. In **Fig. 2b**, all three independent experiments were performed with the same isolation of murine BMDMs whereas data generated in other figures were the BMDMs from two different isolations.

In panels b) and h) all individual values are not visible. Was h) done only 2x? Dotted lines connecting associated values are difficult to track (especially in g) and h)).

To better visualize the individual values in **Fig. 2**, we have now changed the original graphs of paired dots connected with lines in the entire paper to the classical bar graph showing individual plots labeled with different colors for data collected from the same independent experiment. We have also included one more independent experiment in **Fig. 2h** to make triplicates.

The effect of *gra45* ko is lost with 5 ng IFN. In Fig. 5c the difference between wt and *gra45* ko increases with increasing IFN concentrations (the difference is best at 50 ng IFN). This requires an explanation.

We respectfully disagree with the reviewer that the effect of $\Delta gra45$ parasites is lost with 5 ng/mL of IFN γ in the original **Fig. 2b**. $\Delta gra45$ parasites still have decreased (but not significantly) fitness in murine BMDMs pre-stimulated with 5 ng/mL of IFN γ compared to the WT parasites. As we mentioned above, we isolated two different batches of murine BMDMs, and all three independent experiments presented in the original **Fig. 2b** were performed from the first isolation whereas the other four independent experiments were tested with another (2nd) isolation. We have tested again the growth of $\Delta gra45$ parasites with the 2nd isolation of murine BMDMs with IFN γ stimulation and added this data to the new **Fig. 2b**. With the same statistical analysis, the current data with five independent experiments showed that $\Delta gra45$ parasites have

significantly reduced growth compared to the WT parasites in BMDMs activated with both 1 ng/mL and 5 ng/mL of IFN γ .

For g) and h) pictures should be provided.

We determined the ratio of WT parasites (GFP-positive) vs. knockout parasites (GFP-negative) by counting the number of GFP-positive plaques and GFP-negative plaques directly under the fluorescence microscope with 4x objective. Unfortunately, we did not capture any pictures for **Fig. 2g** and **2h**. Regardless we do not believe that pictures would add interesting information to this manuscript.

Plaque assay in BMDM and HFF is missing in M&M.

We have provided the plaque assay in HFFs in the methods section from line 687 to 689. However, we did not perform plaque assays in BMDMs for this study.

Fig. 3:

What is the evidence that TGGT1_263560 is in fact a putative GRA protein?

We have now provided the data showing the co-localization of TGGT1_263560 with GRA2 in intracellular parasites. The data is added to the **Supplementary Fig. 5i**.

Suppl. Fig. 3: In panel f) the complemented *gra45* ko strain was done only 3 times (all the other strains have been done 6 times). Please show only the corresponding values (3 in every case) for all the strains here. Alternative: Repeat the experiment with the same number of replicates.

We accepted the reviewer's suggestion and repeated the experiment. The three times paired data for all the tested strains have been included in the new **Supplementary Fig. 5f**.

Suppl. Fig. 4:

a) Lane 206/261 (GRA45:GRA44 interaction). Reference is missing.

We have added the reference.

Why didn't the authors investigate localisation/aggregation of GRA44 in the absence of GRA45? In my view, this would be an informative experiment because an interaction between GRA45 and GRA44 has already been demonstrated.

We thank the reviewer for the suggestion to investigate the localization of GRA44 in Δ *gra45* parasites, which was indeed an informative experiment supporting our hypothesis. To test the localization, a C-terminal Myc-tagged GRA44 construct was transiently transfected into the WT, Δ *gra45*, and Δ *gra45* + *GRA45HA* parasites followed by fixing at 20 h post-infection and staining with antibodies against Myc epitope. We have observed that a significantly larger fraction of GRA44 remained inside the cytosol of Δ *gra45* parasites compared to WT and Δ *gra45* + *GRA45HA* parasites (**Fig. 4h**). The result observed in Δ *gra45* parasites is clearly different from the localization of other tested GRAs, which are still secreted into the PV but not going to the PVM. In this study, GRA44 is the only known interaction partner of GRA45 and does not contain a predicted hydrophobic domain suggesting GRA45 probably directs the trafficking of GRA44 in the early secretory pathway, which is evidence supporting the chaperone-like function of GRA45.

No indication for structural homology in Suppl. Fig. 4a. Please add.

The conserved secondary structures were indicated with red lines for helix and blue lines for strand in the new **Supplementary Fig. 7a** (we have moved the initial Supplementary Fig. 4 to Supplementary Fig. 7).

The α -Crystallin domain indicated with a red box has high structural homology between the two bacterial sHSPs and GRA45 as well as its orthologs.

Amino acid sequence for at least one HSP is missing in Suppl. Fig. 4a. Please add.

We have added HSP20 from *Xylella fastidiosa* and ApgA from *Salmonella typhimurium* into the alignment and presented the data in the new **Supplementary Fig. 7a**. In addition, we have indicated the homologous α -Crystallin domain and four sHSP motifs (VKV from amino acid 139 to 141, VEV from amino acid 162 to 164, IDV from amino acid 205 to 207 and amino acid 291 to 293) in the new figure. As mentioned in the manuscript from line 251 and 252, the primary sequence search of GRA45 failed to provide any homology region or known domains. The homology to α -Crystallin domain of sHSPs from various species was predicted based on secondary or higher structure modeling of GRA45. Thus, it is not surprising that the predicted α -Crystallin domain or sHSP motifs differ significantly between GRA45 and the two sHSPs.

b) In case of intracellular parasites, no PVM localization is visible (in case of the proteins for that PVM localization has been described). Why?

We have used a different permeabilization protocol (0.2% Saponin for 10 mins) and obtained better quality images of PVM-localized GRA5 and GRA7. The new data are now in the new **Supplementary Fig. 7b**.

The merged pictures have a bad contrast and are therefore difficult to evaluate. Please fix.

We have modified the merged pictures with only showing fluorescent channels and provided the corresponding brightfield images in the new **Supplementary Fig. 7b**.

Fig. 4:

a) As starting point of the study, the authors used a genome-wide-loss-of-function screen in murine macrophages. At the end, the contribution of GRA45 to virulence was investigated in vivo (mice). Why were all in vitro experiments in between done in human (HFF) cells? In my view, this switch of cell type is difficult to retrace (even if the mechanism/role of GRA45 could be very well conserved in mouse and human cells). Murine fibroblasts are certainly appropriate for all the experiments that were performed in HFF cells.

We agree that we could have provided a better rationale for switching of the cell types for verifying the chaperone-like function of GRA45. However, we believe that HFFs were appropriate to further investigate the function of GRA45 because: a) the effect of GRA45 on parasite growth and survival in response to IFN γ -induced restriction is not only observed in murine macrophages but also in rat and human macrophages, thus using human cells seems appropriate; b) our data from extracellular parasites (**Fig. 4a** and **4b**) indicate that GRA45 performs its chaperone-like function inside the parasite where GRA45 maintains the solubility of the GRAs tested in our study; c) the localization of GRAs tested in our study (GRA5, GRA7, MAF1, GRA23, GRA44, GRA16, and GRA24) are unlikely to differ in cells from different host species. So although we could perform all these experiments in murine fibroblasts where we will have an extremely high chance to show the same results that we obtained in HFFs. However, it is unclear to us what we would be able to conclude from that besides knowing murine fibroblasts behave similarly to HFFs.

The intensity of single bands is very inconsistent between single panels:

Why is the GRA2 monomer much more prominent in the NP-40/Tx-100 conditions compared to the PBS condition? If these different intensities reflect different efficiencies of cell lysis, why is the same difference not visible in case of the GRA7 or SAG1 monomers?

The intensity of single bands is very inconsistent within single panels:

Why is the intensity of the GRA2 monomer in the PBS condition very different in wt compared to Δ gra45+GRA45HA? Why is the GRA2 monomer in the wt strain exclusively in the pellet and in case of Δ gra45+GRA45HA exclusively in the supernatant?

Why is the intensity of the high molecular weight complex dramatically different in wt compared to $\Delta gra45$? The explanation here might be that these “aggregates” contain a lot of different mislocalised or misregulated (GRA) proteins that are unspecifically recognised by the detection antibody. But what is then the exact proportion of the respective protein (GRA2 or GRA7) within this complex? It is impossible to say. Why is the high molecular weight complex in the NP-40 condition in case of $\Delta gra45$ +GRA45HA much more prominent in the supernatant compared to the wt strain (in this case almost nothing is visible in the supernatant)?

If I compare $\Delta gra45$ and $\Delta gra45$ +GRA45HA, the intensity of the GRA2 monomer is very different in NP-40 condition but not in the Tx-100 condition. Why? This difference is not visible for the GRA7 monomer. Here, all conditions and strains show the same intensity of the GRA7 monomer. Why?

We understand the concerns from the reviewer on the quality of the GRA2 blot. To solve this, we repeated the experiments three more times and have now added a better image for GRA2 as the new **Fig. 4a**. However, we only did the NP-40 extraction in the additional three individual experiments because a previous study showed the 1% NP-40 is sufficient for solubilizing the GRA aggregates (Gendrin et al, *Traffic*, 2008). To know the solubility of GRA2 and GRA7 from different extraction, we quantified the intensity of bands corresponding to their monomer and high molecular weight (MW) and determined the percentage of insoluble fraction in the total fractions (**Supplementary Fig. 8a and 8b**). Compared to WT and $\Delta gra45$ + GRA45HA parasites, $\Delta gra45$ parasites presented more high MW bands of GRA2 and GRA7 in the pellet fraction from both PBS and NP-40 extractions. However, due to the variability of the intensities between experiments only insoluble GRA2 high MW bands were significantly more present in the $\Delta gra45$ parasite compared to the WT parasite.

We respectfully disagree with the reviewer that the high MW bands are unspecifically recognized by the detection antibodies. A previous study (Labruyere et al, *Mol Biochem Parasitol*, 1999) suggested GRAs form various high MW complexes, which can be detected using specific primary antibodies against GRA2, GRA4 or GRA6. We believe the high MW bands we observed are some sort of complexes. However, the reason that the intensity of this insoluble high MW complex is much higher in $\Delta gra45$ parasites could be because without GRA45 other GRAs such as GRA2 and GRA7 form aggregates as GRA45 has a chaperone-like function, which is important for the prevention of protein aggregates. The whole purpose of these Western blots was to show that without GRA45 the other GRAs form aggregates that can not be easily solubilized. If the aggregates are less soluble there would be less normal GRA2 and GRA7 present as monomers. Somehow the term “aggregation” caused confusion. Thus, we have modified our description for this result to “less solubilization of the high MW bands was observed in the $\Delta gra45$ parasites” instead of saying “GRA45 prevents the aggregation of GRAs”.

In addition, it should also be noted that our data indicate that GRA45 plays a role in the correct trafficking of other GRAs in the secretory pathway as mutations in GRA45 led to GRA45 and with it other GRAs to get partially stuck in early secretory organelles (**Fig. 5b and 5c**). If GRA45 shields the hydrophobic domain of other GRAs to prevent their insertion in the ER membrane then without GRA45 these GRAs might get inserted into ER membrane. This could also affect if these GRAs end up in pellet or supernatant fractions.

b) The quality of the Western blots presented here is quite poor. Bands are only partially visible or divided into pieces (e.g. GRA1 monomers). Some signals are heavily distorted (e.g. GRA5 monomers). Why do the authors strip the membranes between each blot instead of running/staining individual membranes with the individual antibodies? A loading control is missing and should be added. ROP5 (about 60 kDa) could be an option that would not interfere with high molecular weight complexes or monomers. The anti ROP5 antibody was used by the authors before and should therefore be available.

Besides these reservations some inconsistencies should be clarified:

The LSS – as the authors state – contains PV membranes and soluble material. Why is GRA7 (a PVM protein) not visible in the LSS (at least not in case of the wt strain)?

For GRA2 and GRA5 monomers, the pattern is exactly the same for all the strains used here. Why?

GRA1 WT: Why does the HSS-PBS clearly shows a signal for the monomer but the LSS is completely blank (if I understood correctly the HSS-PBS and HSP-PBS are made from the LSS)? GRA1 is a soluble protein. In that case, why is the GRA1 not present in the LSS anyway? Why does Tx-100 treatment of the HSP-PBS results in a strong band that is absent in the HSP-PBS?

GRA1 Δ gra45: GRA1 is a soluble protein and absence of GRA45 leads to its aggregation. Why is then the LSS monomer so prominent (in comparison to the wt strain)?

Did the authors not properly adjust the volumes? This is a must for these experiments.

To give the answers to the reviewer's concern, we have first blotted SAG1 for these samples as a control for checking the efficiency of PVM fraction separated from intact parasites with high-speed centrifugation. However, we found that SAG1 also presented in the HSP/HSS fraction, which was supposed to be only the PVM fractions, suggesting the poor preparation of the PVM fractions in these samples. Due to this, we have removed the data for intracellular parasites from the manuscript as the new **Fig. 4a** and **4b** provided evidence that in extracellular Δ gra45 parasites GRAs are present in high MW bands that are less solubilized by detergent (likely aggregates). Furthermore, our new data from **Fig. 4c - 4f** also showed that transmembrane domain-containing GRAs were mislocalized in Δ gra45 parasites. In addition, the new data from **Fig. 5** showed that in GRA45 mutants that are predicted to affect its chaperone function the localization of GRA45 and GRA7 is affected indicating that GRA45 plays a role in the correct trafficking of other GRAs in the secretory pathway. We believe these data are sufficient to support our hypothesis that GRA45 plays a chaperone-like function.

The authors should demonstrate localisation/mislocalisation of all the GRA proteins investigated here by immunofluorescence microscopy in addition (like they did for GRA23, GRA16 and GRA24). Especially for GRA7 monomers, the only difference is the intensity of the phosphorylated form. From this the authors conclude that GRA7 is mislocalised in the absence of GRA45. Is GRA7 exclusively phosphorylated in infected cells (like the authors claim)? In Fig. 4a (using extracellular tachyzoites) one gets the impression that GRA7 also appears at different molecular weights (not clearly visible because the authors show an overexposed membrane). Therefore, the localisation/mislocalisation of GRA7 has to be addressed by an alternative approach (e.g. immunofluorescence).

We thank and accept the reviewer's suggestion to investigate the localization of GRA7 by immunofluorescence assay. In addition to GRA7, we have observed and quantified the localization of other PVM-integrated or -associated GRAs (such as GRA5, MAF1, and GRA23). The data included in the new **Fig. 4c - 4f** showed that parasites lacking GRA45 indeed have significantly reduced PVM localization of all these tested GRAs.

To confirm the conclusion that GRA45 is a general regulator of other GRA proteins (prevents other GRA proteins from aggregation), GRA45 interaction with different GRA proteins (GRA1, GRA2, GRA5 and GRA7) has to be investigated.

As we explained above, the interaction experiments were not performed due to scientific and technical challenges. However, we believe that the reviewer's suggested experiments presented in the new **Fig. 4h** provide an answer to this. As the known interaction partner of GRA45, GRA44 was significantly retained inside the cytosol of Δ gra45 parasites instead of secreted to the PV (**Fig. 4h**) revealing that the trafficking of GRA44 in the early secretory pathway needs the interaction with GRA45. Thus, it is most likely that GRA45 also interacts with other GRAs during their trafficking and secretion. Because we did not provide the interaction of GRA45 with other PVM-localized GRAs, we have rephrased the description of the chaperone-like function for GRA45 instead of stating GRA45 as a general regulator.

g), i) In my opinion, normalized control data violate all the assumptions of the one-way ANOVA. Can the authors please comment on that issue? Why did they decide to set the control at 100%?

The nuclear translocation of GRA16/24 was observed in the parasites transiently transfected with ectopic expression vectors containing GRA16/24 fused with the HA epitope. Thus, the raw intensity of GRA16/24 in host nuclei may vary between the independent experiments due to several factors, such as transfection efficiency, the copy number of transfected plasmids, and immunofluorescent procedure. However, to eliminate the reviewer's concern on this, we have provided the raw average intensity of GRA16/24 in the new **Fig. 4i** and **4j**.

f), h) I don't see any aggregation/mislocalisation of GRA16 (f) or GRA24 (h) in the absence of GRA45 in the pictures provided here.

The GRA16/24 staining in **host nuclei** indeed disappeared in $\Delta gra45$ parasites similar to what we observed for $\Delta myr1$ parasites. Maybe we did not explain well that the phenotype of absent GRA16/24 translocation is likely the consequence of a deficient/mislocalized MYR complex in the PV of $\Delta gra45$ parasites. Likely because our data indicate that GRA45 is needed to maintain the solubility and correct localization of potential PVM translocon components (for example MYR1 in **Supplementary Fig. 8c**). Therefore, GRA45 deletion will likely cause PVM translocon component mislocalization resulting in the abolishment of GRA16/24 export beyond the PVM. In addition, both GRA16/24 harbor large intrinsically disordered regions combined with short linear motifs (Hakimi and Bougdour, *Curr Opin Microbiol*, 2015), which likely provide the structural flexibility for maintaining their solubility when they are inside dense granules as well as in the PV. Thus, it is in our expectation that no aggregation/mislocalization of GRA16/24 inside the **PV** should be observed in $\Delta gra45$ parasites. Up to date, since no study has dissected the structure of the *Toxoplasma* PVM-localized translocon, it is likely that one or more components of the MYR complex function as chaperone-like protein involved in the trafficking of exported GRAs in the secretory pathway. Therefore, we cannot exclude the possibility that GRA45 synergizes with other MYR members to help GRA16/24 reach the PVM.

Fig. 5:

The significant *in vivo* phenotype for $\Delta gra45$ is not reflected by all the *in vitro* data. Mislocalisation of GRA proteins in absence of GRA45 should be reflected by GBP and IRG (GRA7 does influence Irga6) loading to the PVM. The authors could determine if numbers of vacuoles positive for at least Irga6 change in the absence of GRA45.

According to the reviewer's suggestion, we have observed the percentage of IRGA6-positive and IRGB6-positive vacuoles in IFN γ -activated murine BMDMs infected with WT, $\Delta gra45$ or $\Delta gra45+GRA45HA$ parasites. However, the result included in the new **Fig. 3a** and **3b** showed that $\Delta gra45$ parasites have a similar level of IRGA6 and IRGB6 coating like the WT parasites. It is worth to notice that although GRA7 promotes rapid turnover of the IRGA6 oligomer, the coating of IRGA6 and IRGB6 only increased in $\Delta gra7\Delta drop18$ parasites but not in $\Delta gra7$ parasites (Alaganan et al, *Proc Natl Acad Sci USA*, 2014). Given the data that only a portion of GRA7 mislocalized in $\Delta gra45$ parasites (**Fig. 4d**) combined with the fact GRA45 is also required for parasite growth in IFN γ -activated THP-1 macrophages (human cells do not have IRGs), it is reasonable to believe that the IFN γ resistance is due to the combination of various GRAs of which the correct trafficking to their final destination (e.g, PVM or host cytoplasm) is dependent on GRA45. In addition to testing $\Delta gra45$ parasites, we have observed the loading of IRGA6 and IRGB6 to the vacuole of $\Delta gra22$, $\Delta TGGT1_263560$, and $\Delta TGGT1_269950$ parasites, and these data are now included in the new **Fig. 3a** and **3b**.

However, we respectfully disagree with the reviewer that the significant *in vivo* phenotype of $\Delta gra45$ is not reflected by all the *in vitro* data. Various studies have shown that IFN γ resistance plays an irreplaceable role in parasite virulence in mice. Our *in vitro* data showed that $\Delta gra45$ parasites, but not $\Delta myr1$ parasites, were susceptible to IFN γ -induced parasite growth inhibition in murine macrophages. Given that macrophages are the major cell type preferentially infected with *Toxoplasma in vivo* (Jensen et al, *Cell Host Microbe*, 2011) and play an essential role in the early immune response against the parasite, the distinct growth phenotype of $\Delta gra45$ and $\Delta myr1$ parasites in IFN γ -activated murine macrophages corresponds to their *in vivo* virulence. In addition, GRA23 was mislocalized in only $\Delta gra45$ parasites but not

$\Delta myr1$ parasites. Thus, it is most likely other PVM-localized GRAs that are mislocalized in the PV of $\Delta gra45$ parasites have a normal localization in $\Delta myr1$ parasites. As some of the PVM-localized GRAs were previously shown to confer the *in vivo* fitness, the abnormal localization of GRAs to the PVM in $\Delta gra45$ parasites also reflects the *in vivo* phenotype of $\Delta gra45$ but not $\Delta myr1$ parasites.

Other comments:

Lane 63: The effect of ROP16 and GRA15 was only tested for GBP1 in that study (34).

Lane 65: The effect of ROP54 was only tested for GBP2 in that study (36).

Lane 66: The effect of GRA12 on IRG loading was only tested for Irgb6 in that study (37)

We thank the reviewer for pointing this out, we have changed these sentences accordingly.

Reviewers' Comments:

Reviewer #1:

Remarks to the Author:

The authors have put significant effort into reanalysing the data, adding extra experiments and simplifying the text and figures. This results in a much improved and streamlined version. The description of the CRISPR screen is much clearer and while not ideal that there are different conditions, the combined analysis provides valuable data and the differences between conditions are discussed. They show the impact of GRA45 on multiple GRA localisations/phenotypes and importantly have added in GRA45 chaperone domain mutants, helping to solidify this aspect of the paper.

Minor comments

1. It would be helpful to have known IFN γ controls marked on Fig 1d. ROP18 and GRA12 are mentioned in the discussion as not having met the criteria but the reader is left wondering about these when initially looking at the data. Or if not, this could be mentioned in relation to 1d with further discussion of the reasons later.
2. The reasoning (Line 269) that as GRA45 is in the PV lumen it can't be directly be affecting IFN γ defences could be specified. I presume you mean unlikely to directly interact with host cell proteins. But there could be direct effects on the PV membrane that change the ability of the host cell to interact with the parasites. I am not saying this is happening, but cannot be excluded.
3. Line 449-450 The authors reference Young et al as showing that MYR1 KO has a virulence defect in the CRISPR screen. This paper actually shows the opposite that there is no MYR1 phenotype observed in the combined CRISPR pool, despite a strong virulence defect in the individual KO.

Reviewer #2:

Remarks to the Author:

The authors have addressed most of my comments and have sufficiently altered the manuscript. The manuscript is much improved and their thoroughness is much appreciated. I just have a few outstanding points:

Fig. 2:

For g) and h) pictures should be provided.

Pictures form the basis for these results and it would be valuable to see some examples. It is not so much about adding new information to the manuscript. It is rather that pictures visualize the depicted results. This is always helpful, especially for those who are not so familiar with plaque assays. This is also the case for all the immunofluorescence experiments. In that case, the authors decided to show exemplary pictures.

Fig. 4:

If – like the authors say – murine fibroblasts behave similarly to HFFs and there is an extremely high chance to show the same results, this would be an argument for not switching the cell type in order to be as consistent as possible. However, I will not put any pressure on the authors to repeat these experiments in murine cells.

The point I wanted to make was expressed misleadingly and I have to apologize for that. The total amounts of GRA2 and GRA7 in the old Western blots are totally different comparing the different strains used here (after adding up the signal intensities of the monomers and high MW complexes). These differences are still visible in the new Western blot for GRA7. Where is all the GRA7 that I see in the high MW complex with Δ gra45 in case of wt strain infections? In other words: why are the intensities of monomers in wt infections not comparable with the intensities of high MW complexes in ko infections if signals exclusively reflect GRA7 amounts? In my opinion, this indicates unspecific recognition of (GRA) proteins within the high MW GRA7 complex by the anti GRA7 antibody. Assuming that this is true, what is really the GRA7 amount in this complex? Besides, I don't see any differences in GRA2 and GRA7 monomer intensities whatsoever.

Fig. 5:

I appreciate the argument that only a portion of GRA7 is mislocalized in Δ gra45 parasites and the resistance phenotype is probably multifactorial. However, I don't understand the argument "...the coating of IRGA6 and IRGB6 only increased in Δ gra7 Δ rop18 parasites but not in Δ gra7 parasites (Alaganan et al, Proc Natl Acad Sci USA, 2014)". In Hermanns et al, 2016, IRGA6 loading (as well as all other IRGs tested in that study) is significantly increased with Δ gra7 parasites. If GRA45 has a significant impact on GRA7 localization this should be reflected by vacuolar IRGA6 loading. The authors could determine the IRGA6 (and IRGB6) amount at the PVM in addition.

Reviewer #1 (Remarks to the Author):

The authors have put significant effort into reanalysing the data, adding extra experiments and simplifying the text and figures. This results in a much improved and streamlined version. The description of the CRISPR screen is much clearer and while not ideal that there are different conditions, the combined analysis provides valuable data and the differences between conditions are discussed. They show the impact of GRA45 on multiple GRA localisations/phenotypes and importantly have added in GRA45 chaperone domain mutants, helping to solidify this aspect of the paper.

We appreciate the reviewer's suggestions on how to improve the manuscript.

Minor comments

1. It would be helpful to have known IFN γ controls marked on Fig 1d. ROP18 and GRA12 are mentioned in the discussion as not having met the criteria but the reader is left wondering about these when initially looking at the data. Or if not, this could be mentioned in relation to 1d with further discussion of the reasons later.

We have mentioned ROP18 and GRA12 in the result section related to Fig. 1d from line 164 to 167. In addition, we have discussed these two proteins further in the discussion from line 508 to 515.

2. The reasoning (Line 269) that as GRA45 is in the PV lumen it can't be directly be affecting IFN γ defences could be specified. I presume you mean unlikely to directly interact with host cell proteins. But there could be direct effects on the PV membrane that change the ability of the host cell to interact with the parasites. I am not saying this is happening, but cannot be excluded.

We thank the reviewer for pointing this out and have modified the text accordingly.

3. Line 449-450 The authors reference Young et al as showing that MYR1 KO has a virulence defect in the CRISPR screen. This paper actually shows the opposite that there is no MYR1 phenotype observed in the combined CRISPR pool, despite a strong virulence defect in the individual KO.

We apologize for the mistake and have corrected this in the revised manuscript from line 456 to 458.

Reviewer #2 (Remarks to the Author):

The authors have addressed most of my comments and have sufficiently altered the manuscript. The manuscript is much improved and their thoroughness is much appreciated.

We appreciate the reviewer's suggestions on how to improve the manuscript.

I just have a few outstanding points:

Fig. 2:

For g) and h) pictures should be provided.

Pictures form the basis for these results and it would be valuable to see some examples. It is not so much about adding new information to the manuscript. It is rather that pictures visualize the depicted results. This is always helpful, especially for those who are not so familiar with plaque assays. This is also the case for all the immunofluorescence experiments. In that case, the authors decided to show exemplary pictures. [I think that here we should not go into detail; I will maybe discuss this with the editor because it doesn't make sense to do a separate experiment for this]

We understand the reviewer's concern on this point, however, as we replied previously, these plaque assays were scored for green (GFP-positive) and non-green (GFP-negative) plaques and no pictures were taken. The plaques themselves look the same as the other pictures of plaques in previous publications from our group and other groups, with the difference that some are green and others not. To provide these pictures, we would have to re-order mice, isolate macrophages, and redo the growth competition assay. Currently, our mouse facility at the university is not allowed to order any mice due to the COVID-19 pandemic. Also, we feel that the addition of these pictures will not change or improve any of the data or conclusions of our manuscript.

Fig. 4:

If – like the authors say – murine fibroblasts behave similarly to HFFs and there is an extremely high chance to show the same results, this would be an argument for not switching the cell type in order to be as consistent as possible. However, I will not put any pressure on the authors to repeat these experiments in murine cells.

We thank the reviewer for understanding that during these times it will be difficult to repeat all these experiments.

The point I wanted to make was expressed misleadingly and I have to apologize for that. The total amounts of GRA2 and GRA7 in the old Western blots are totally different comparing the different strains used here (after adding up the signal intensities of the monomers and high MW complexes). These differences are still visible in the new Western blot for GRA7. Where is all the GRA7 that I see in the high MW complex with Δ gra45 in case of wt strain infections? In other words: why are the intensities of monomers in wt infections not comparable with the intensities of high MW complexes in ko infections if signals exclusively reflect GRA7 amounts? In my opinion, this indicates unspecific recognition of (GRA) proteins within the high MW GRA7 complex by the anti GRA7 antibody. Assuming that this is true, what is really the GRA7 amount in this complex?

Besides, I don't see any differences in GRA2 and GRA7 monomer intensities whatsoever.

We thank the reviewer for clearing the confusion and we have now understood the reviewer's concern on the specificity of anti-GRA7 antibodies and the unequal total amount of GRA7 between different strains. To test the specificity of the antibodies, cell fractionation was performed in PBS-resuspended Δ gra7 parasites along with wild-type parasites according to the methods described in the manuscript and both pellet and supernatant fraction were blotted with anti-GRA7 antibodies used in this study. The results shown in the new **Supplementary Fig. 8c**, show that high molecular weight (HWM) bands and monomeric GRA7 were

only observed in the fractions from wild-type parasites but not $\Delta gra7$ parasites. This demonstrates that the HMW bands detected with the anti-GRA7 antibody is not due to aspecific detection of other GRAs. In addition, we now provide better images in **Fig. 4b** with an approximately equivalent total amount of GRA7 between wild-type and $\Delta gra45$ parasites.

The main goal of the Western blot experiment was to investigate if $\Delta gra45$ parasites have more insoluble GRA aggregates compared to wild-type and complemented parasites. The insolubility of aggregated GRAs is determined by the presence of high molecular weight (HMW) bands that persist even after NP40 treatment. Thus, it was our expectation that no differences in the intensities of monomeric GRA2 and GRA7 were observed between strains.

Fig. 5:

I appreciate the argument that only a portion of GRA7 is mislocalized in $\Delta gra45$ parasites and the resistance phenotype is probably multifactorial. However, I don't understand the argument "...the coating of IRGA6 and IRGB6 only increased in $\Delta gra7\Delta rop18$ parasites but not in $\Delta gra7$ parasites (Alaganan et al, *Proc Natl Acad Sci USA*, 2014)". In Hermanns et al, 2016, IRGA6 loading (as well as all other IRGs tested in that study) is significantly increased with $\Delta gra7$ parasites. If GRA45 has a significant impact on GRA7 localization this should be reflected by vacuolar IRGA6 loading. The authors could determine the IRGA6 (and IRGB6) amount at the PVM in addition.

We thank the reviewer for this suggestion and have determined the intensity of IRGA6 and IRGB6 coated on the vacuole of WT or $\Delta gra45$ parasites according to the study mentioned by the reviewer (Hermanns et al., *Cell Microbiol*, 2016). However, the result showed that the intensity of IRGA6 and IRGB6 from three independent experiments are similar between WT and $\Delta gra45$ parasites (as shown in the below figure). Although the study we previously mentioned (Alaganan et al, *Proc Natl Acad Sci USA*, 2014) did not quantify the intensity of IRGA6 and IRGB6 coated on the vacuole of $\Delta gra7$ parasites, our results in **Fig. 3a** and **3b** are similar to Fig. 3A to 3D from this study as they described "Loss of GRA7 did not directly affect IRG loading or clearance in IFN- γ -activated macrophages". We believe these paradoxical results are due to: a) the differences in cell types, as both Alaganan et al, *Proc Natl Acad Sci USA*, 2014 and our study were carried out in murine macrophages whereas mouse embryonic fibroblasts (MEFs) were used in Hermanns et al., *Cell Microbiol*, 2016; b) GRA7 is not the only PVM-localized GRA affected in $\Delta gra45$ parasites, it is possible that GRA45 plays a role in the trafficking and localization of other GRAs maintaining lipid composition of the PVM. The specific phospholipids present on the PVM are known to influence IRGB6 recruitment (Lee et al, *Life Sci Alliance*, 2020).